**Data Availability Statement:** The data underlying the results presented in the study are available

# Evaluating changes in the prevalence of female genital mutilation/cutting among 0-14 years old girls in Nigeria using data from multiple surveys: A novel Bayesian hierarchical spatio-temporal model

Chibuzor Christopher Nnanatu[1,2]*, Glory Atilola[1], Paul Komba[1], Lubanzadio Mavatikua[1], Zhuzhi Moore[3], Dennis Matanda[4], Otibho Obianwu[5], Ngianga-Bakwin Kandala[6,7]

1 Northumbria University, Newcastle, United Kingdom, 2 Nnamdi Azikiwe University, Awka, Nigeria, 3 Independent Consultant, Seattle, Washington, United States of America, 4 Population Council, Nairobi, Kenya, 5 Population Council, Abuja, Nigeria, 6 Division of Epidemiology and Biostatistics, School of Public Health, University of The Witwatersrand, Johannesburg, South Africa, 7 Division of Health Sciences, Warwick Medical School, University of Warwick, Coventry, United Kingdom

* chibuzor.nnanatu@northumbria.ac.uk

## Abstract

Female genital mutilation/cutting (FGM/C) is considered a public health and human rights concern, mainly concentrated in Africa, and has been targeted for elimination under the sustainable development goals. Interventions aimed at ending the practice often rely on data from household surveys which employ complex designs leading to outcomes that are not totally independent, thus requiring advanced statistical techniques. Combining data from multiple surveys within robust statistical framework holds promise to provide more precise estimates due to increased sample size, and accurately identify 'hotspots' and allow for assessment of changes over time. In this study, rich datasets from six (6) successive waves of the Nigeria Demographic and Health Surveys and Multiple Indicator Cluster Surveys undertaken between 2003 and 2016/17, were combined and analyzed in order to better assess changes in the likelihood and prevalence of FGM/C among 0-14-year old girls in Nigeria. We used Bayesian hierarchical regression models which explicitly accounted for the inherent spatial and temporal autocorrelations within the data while simultaneously adjusting for variations due to different survey methods and the effects of linear and non-linear covariates. Parameters were estimated using Markov chain Mote Carlo techniques and model fit assessments were based on Deviance Information Criterion. Results show that prevalence of FGM/C among 0–14 years old girls in Nigeria varied over time and across geographical locations and peaked in 2008 with a shift from South to North. A girl was more likely to be cut if her mother was cut, supported FGM/C continuation, or had no higher education. The effects of mother's age, wealth and type of residence (urban-rural) were no longer significant in 2016. These results reflect the gains of interventions over the years, but

from (https://dhsprogram.com/ and http://mics.unicef.org/).

**Funding:** Funding for this work was provided by the UK Aid and the UK Government through the Department for International Development funded project, "Evidence to End FGM/C: Research to Help Girls and Women Thrive," coordinated by Population Council. The funders had no role in study design, data collection and analysis, decision to publish, or preparation of the manuscript.

**Competing interests:** The authors have declared that no competing interests exist.

also echo the belief that FGM/C is a social norm thus requiring tailored all-inclusive interventions for the total abandonment of FGM/C in Nigeria.

## Introduction

Female genital mutilation/cutting (FGM/C) is recognised globally as a violation of the fundamental human rights of girls and women, which has no medical basis and could lead to severe health problems including increased risk of new born deaths [1]. It is estimated that about 200 million girls and women alive today from over 30 countries mainly in Africa, the Middle East and Asia have been subjected to FGM/C and that approximately 3 million young girls are at risk of being cut each year [2]. Consequences of the practice, which is mostly carried out on young girls under the age of 15 years, are well documented in the literature and range from shock to haemorrhage, and from difficulty in passing urine to inhibited orgasm [3–9].

In relation to Nigeria, a report by 28 Too Many in 2018 stated that about 20 million Nigerian women and girls have experienced FGM/C [10]. Also, a recent study by Kandala et al estimated that the national prevalence of FGM/C among women (ages 15–49) stood at 18.4% in 2016/17 representing a decline of 11.2% from 29.6% in 2008, while the national prevalence of FGM/C among girls aged 0–14 years in Nigeria stood at 25.3% in 2016/17, representing a decline of 4.7% from 30.0% in 2008 [11]. This implies that while there has been a sharp decline in prevalence among women aged 15–49 years, the practice seems to still hold sway over 0–14 years old girls in Nigeria despite several concerted efforts aimed at the total eradication of the practice.

In general, efforts to accelerate the abandonment of FGM/C in Nigeria have been a mix of legal /policy and advocacy interventions [12]. In response to the international calls and in line with the Sustainable Development Goals (SGDs) Target 5.3 aimed at the eradication of all forms of harmful practices against women and children including FGM/C by the year 2030, the Nigerian government has passed a federal legislation, the Violence against Persons (Prohibition) Act 2015 (VAPP Act), which strongly prohibits FGM/C and other forms of gender-based violence in Nigeria with provisions including the prosecution of the perpetrators and re-integration of victims into society [10, 13–14]. This, in addition to other national policies such as the 2013/2017 National Policy and Plan for Action for Elimination of FGM/C in Nigeria [15, 16] are initiatives aimed at bringing the practice of FGM/C in Nigeria to an end. In the same spirit, the civil society organizations (CSOs) have also continued to mobilize people against FGM/C through public awareness by forming a partnership with media and the civil society including traditional and religious leaders to disseminate anti-FGM/C messages at federal and state levels with the aim of turning cutters into anti-FGM/C campaigners [17, 18].

Nevertheless, it should be noted that the VAPP Act has hardly been enforced in Nigeria [19] and its implementation varies across the 36 states of the country with some states (especially the high prevalent ones) yet to follow suit [10, 13]. In addition, where it exists, it is even harder to implement the anti-FGM/C law where there is limited presence of law enforcement agents, for example, in the rural areas. Also, the rate of reporting has been low because the perpetrators are almost always family members and it is even possible that the law enforcement agents may sometimes discharge such reports as a family or community matter aimed at preserving one's socio-cultural norms and decide not to meddle [1].

For the reasons of the variations in the level of commitments in stemming the practice of FGM/C at state level in Nigeria, and the compelling belief that FGM/C is deeply-rooted in cultural and social norms [20–23], it then becomes apparent that the need to further investigate trends in the practice as well as the roles of the underlying state structures and the effects of

socio-cultural norms in the persistence of the practice in Nigeria especially among 0–14 years old girls using quality data can never be overemphasized [24]. However, such data are mainly provided by household surveys, namely, the Demographic and Health Surveys (DHS) and the Multiple Indicator Cluster Surveys (MICS), which employ complex designs that are not completely independent thus necessitating the use of advanced statistical techniques.

Bayesian hierarchical regression models which explicitly account for the inherent spatial and temporal autocorrelation within data, while simultaneously controlling for other linear and non-linear effects in a statistically robust framework are popular in the areas of disease mapping and ecological studies, for identifying high risk geographical locations [25, 26]. To date, only a few studies have explored this powerful statistical tool in the context of FGM/C [27–29]. However, these studies have largely utilised data from only one single survey thereby providing only a 'snapshot' assessment. Combining data from multiple surveys within robust statistical framework holds promise to provide more precise estimates, narrower width of the credible interval of estimates and smaller standard errors due to increased sample size. We could then exploit the merits of each survey and more accurately identify 'hotspots' where the practice is still rife and assess changes over time.

The overarching aim of this study, therefore, is to assess how the prevalence and likelihood of female genital mutilation/cutting has changed over time with respect to the roles of socio-cultural norms (operationalized in terms of a woman's FGM/C status and her support for the continuation of the practice), a girl's geographical location and other key determinants in the persistence of FGM/C among 0-14-year-old girls in Nigeria, and identify spatial patterns and 'hotspots'. We combined rich datasets from six (6) successive waves of the Nigeria Demographic and Health Surveys (DHS) and Multiple Indicator Cluster Surveys (MICS) undertaken between 2003 and 2016 using Bayesian hierarchical regression models which explicitly accounted for the inherent spatial and temporal autocorrelations within the data while simultaneously adjusting for variations due to different survey methods and the effects of linear and non-linear covariates. The different years datasets were combined to simultaneously investigate spatial and temporal trends in the practice of female genital mutilation/cutting (FGM/C) across the Nigeria 36 states and the federal capital territory (FCT). In addition, by combining the various datasets, we gained more statistical power to better estimate parameters with higher precision. It is hoped that the statistical evidence generated from the study would serve to facilitate the development and implementation of tailored programmatic interventions that would ensure the total eradication of FGM/C in Nigeria.

Specifically, we seek to answer the following questions:

1. Which individual- and community-level factors are key drivers of FGM/C among 0–14 years old girls in Nigeria?

2. Are there significant effects of a woman's FGM/C status and/or her support for its continuation on the likelihood of her daughter being cut?

3. Is the geographical location (state) in which a girl lives key to her likelihood of experiencing FGM/C?

4. How has the practice of FGM/C among 0–14 years girls varied spatially and temporally in Nigeria?

The remainder of this paper is organised as follows. In Section 2, we give details on the data and statistical methods used in this study and present the results in Section 3. Finally, the key findings are discussed in Section 4.

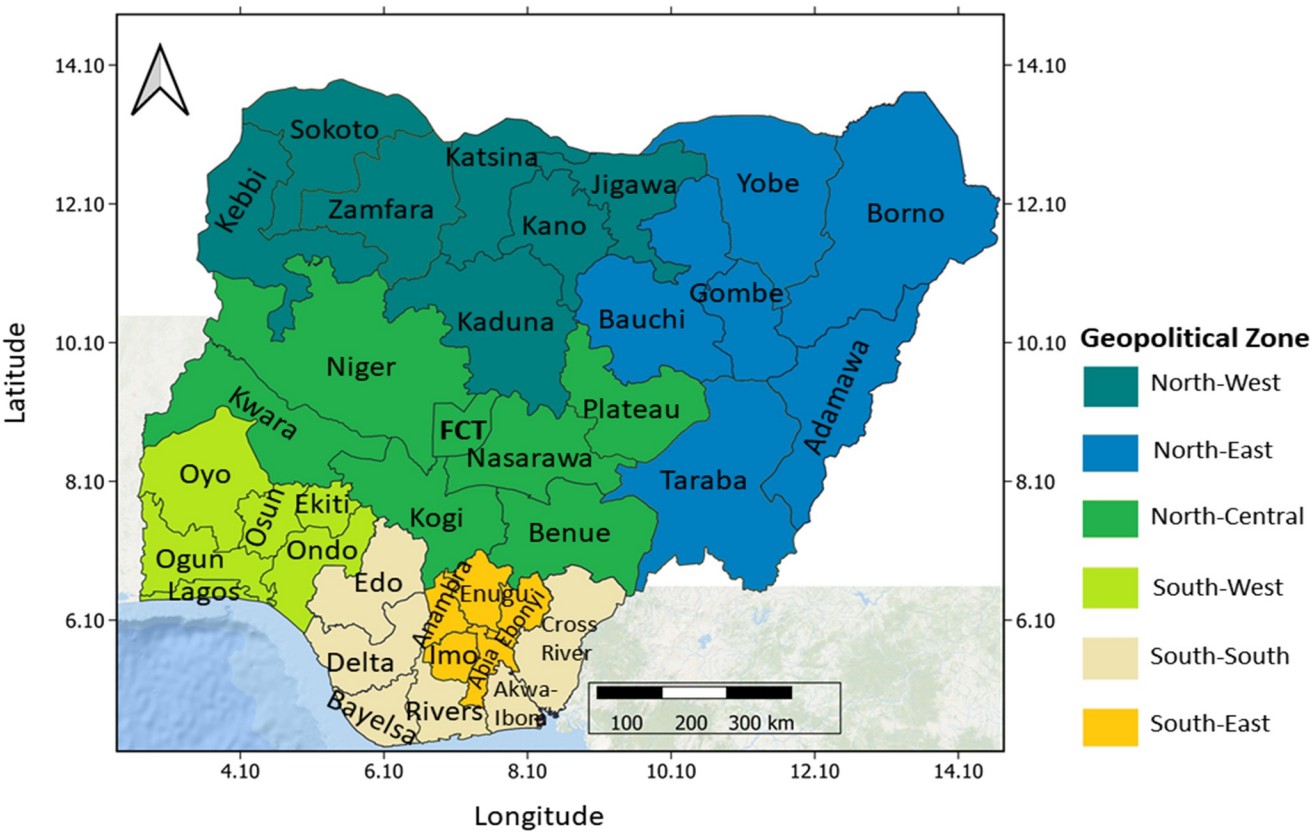

**Fig 1. Map of Nigeria showing the 36 states and the FCT and the six (6) geopolitical zones.** Shapefile republished from DIVA-GIS database (https://www.diva-gis.org/) under a CC BY license, with permission from Global Administrative Areas (GADM), original copyright 2018.

## Materials and methods

### Data sources

Data on FGM/C prevalence were drawn from six nationally representative surveys from Nigeria Demographic and Health Surveys (DHS) and Nigeria Multiple Indicators Cluster Surveys (MICS) comprising the 2003 DHS, 2007 MICS, 2008 DHS, 2011 MICS, 2013 DHS and 2016/17 MICS (henceforth, written as 2016 MICS). Both the DHS and MICS are similarly designed to collect key information including issues around FGM/C from the respondents (women aged 15–49 years) selected from the population of interest using a two-stage, stratified sampling design in all the 36 states in Nigeria and the Federal Capital Territory (FCT) with the clusters as the primary sampling unit (Fig 1). Then, all eligible women of reproductive age (15–49 years) living in the selected households were interviewed. A brief description of the similarities in the survey methods adopted by the DHS and MICS is given in [30]. Also, further details on the study design, sampling, data collection procedures and availability are found in https://dhsprogram.com/ and http://mics.unicef.org/ for DHS and MICS, respectively.

From both the DHS and the MICS datasets, we extracted information on the respondents' daughters aged 0–14 years from the respondents' files and daughters' birth files. In all, the samples consisted of a combined total of 88,319 daughters of 51,141 women who completed the FGM/C modules in the Nigeria DHS and MICS conducted between 2003 and 2016. Fig 2A shows that apart from 2003 DHS, the DHS appear to have a larger coverage than the MICS

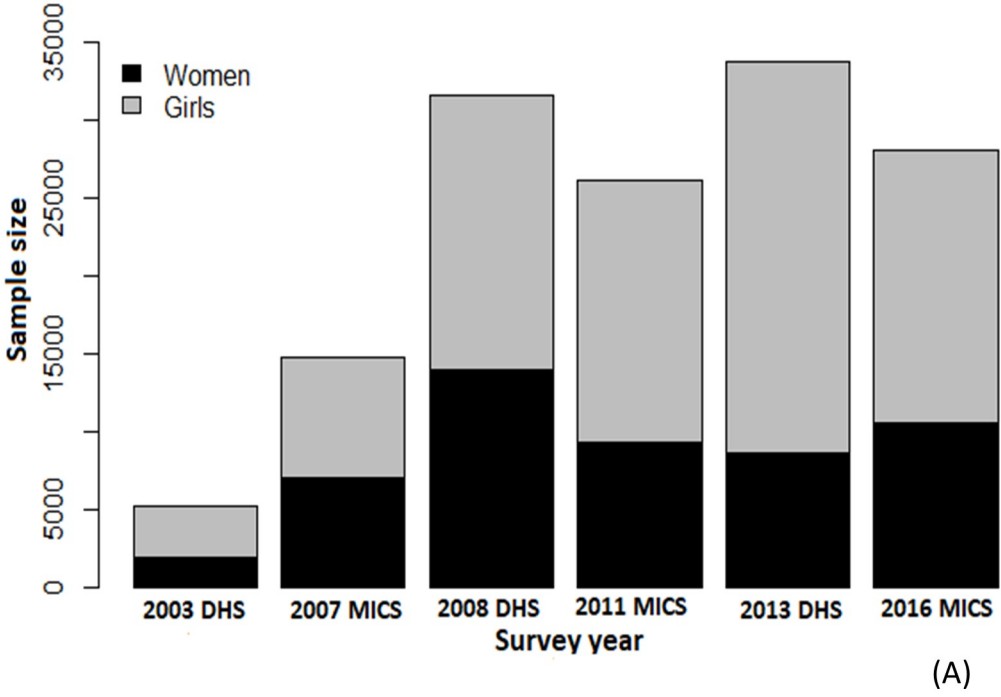

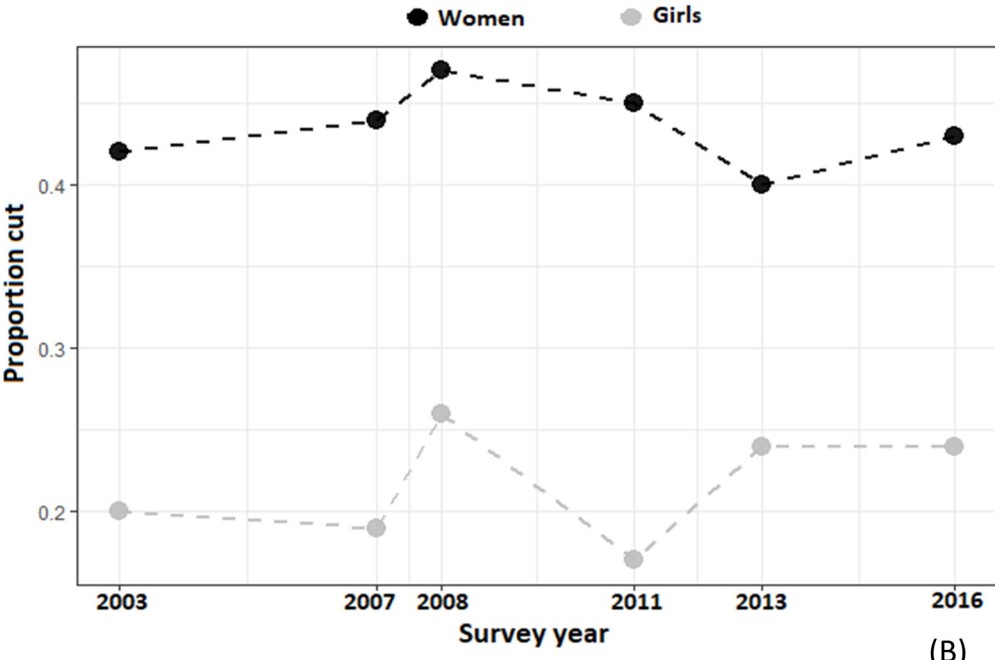

**Fig 2. Data presentation.** (A) Sample size distribution of women (aged 15–49 years) and girls (aged 0–14 year) across the Nigeria DHS and MICS surveys from 2003 to 2016. (B) Proportion of cut women and girls across the datasets.

with consistently higher sample sizes. In addition, the data shows consistently higher propor-tion of cut women than girls across the survey years with a mutual peak in 2008 (Fig 2B).

## Response and exposure variables

For the questions we address in this study, the primary outcome is the FGM/C status of the respondent's daughter aged 0–14 years, that is, whether the respondent's daughter has been cut or not. The outcome is, therefore, a binary variable taking values from the discrete set {0,1} such that a value of 1 indicates that the respondent's daughter has been cut while 0 indicates that the respondent's daughter was not cut. It should be noted, however, that these values only reflected the current status of the respondent's daughter as at the time of the surveys and do not represent their final FGM/C status. This is because a girl who was uncut as at the time data are collected may still be cut in the future.

Furthermore, to investigate the effects of both individual- and community-level covariates required to address the questions, we included measures of socio-demographic factors such as a woman's region and state of residence, type of place of residence (urban vs rural), age, eth-nicity, wealth index, marital status and highest level of education attained (Table 1). Factors of socio-cultural norms are operationalized by a woman's FGM/C status, her support for the con-tinuation of the practice and her beliefs about FGM/C. For the Bayesian regression models, we used state identification numbers ID to capture the unobserved state-level effects of geographi-cal locations, while the type of survey and year are both included to account for the effects of different types of surveys and survey years, respectively (Table 1).

Other variables examined are a girl's current age, girl's age at cutting, type of FGM/C, per-son who performed FGM/C; a woman's religion, belief on the reasons behind the practice of FGM/C; husband/Partner's educational level, household decision making power, and fre-quency of reading newspapers, listening to the radio or watching television.

Finally, for the purposes of examining spatio-temporal variations, two datasets were extracted, namely, the 2016 MICS and a pooled dataset comprising the 2003 to 2016 surveys data combined across common variables.

**Table 1. Variable names, levels and descriptions.**

| Predictor variable | Description | Levels |
|---|---|---|
| Age | Age of the respondent (woman) in years | 15 to 49 |
| Support | Whether respondent supports the continuation of FGM/C. | Stopped, continued, depends/don't know. |
| $fgm_{woman}$ | Respondent's FGM/C status. | Not cut, Cut |
| fgm | FGM/C status of respondent's daughter. | 0-→ Not cut |
| | | 1-→ Cut |
| Wealth | Respondent's household wealth index | Highest, Higher, Middle, Second, Lowest |
| Region | Region of residence | North central, North west, North east, South east, South west, South south |
| Residence | Type of region of residence | Urban, rural |
| Education | Respondent's highest level of education | None, Primary, Secondary, Higher |
| Marital | Marital status | Currently married, formerly married, never married |
| State | Respondents geographical location identification number | 1 to 37 |
| Year | Survey year | 2003, 2007, 2008, 2011, 2013, 2016 |
| Survey | Type of survey | MICS, DHS |

## Statistical analysis

**Bivariate data analysis.** To understand the pairwise association between a girl's FGM/C status and a set of baseline characteristics, we carried out weighted bivariate analyses of both the DHS and MICS datasets. The survey weights allowed us to ensure that our estimates came from samples which were nationally representative. These analyses were conducted in Stata version 14 using the *svy* command ([31]) and weighted outputs were cross tabulated and reported as FGM/C prevalence. Relevant covariates among those that are significantly associated with the response are then included in the Bayesian regression models.

**Test for clustering.** We examined the local and global clustering in the data using Moran's I test to evaluate the clustering pattern of FGM/C among 0–14 years old girls in Nigeria [32, 33]. The key is to determine the geographical locations (states) with significant clustered, dispersed or random structure of the number of girls aged 0–14 years old who have undergone FGM/C. A significantly positive (negative) value of the Moran's I statistics indicates clustered (dispersed) outcome. These analyses were carried out in R statistical programming software version 3.6.1 [34] using *moran.test()* and *moran.mc()* functions available in the **spdep** package in R.

**Bayesian hierarchical spatial and spatio-temporal modelling.** The DHS and MICS data are inherently hierarchical and spatially and temporally autocorrelated largely due to the cluster sampling design adopted by the DHS and MICS. This, in addition to the fact that the values based on a single woman who had at least two children are not independent, justified the need for sophisticated statistical analytical approaches that explicitly allow for autocorrelated response in space and time while simultaneously accounting for linear and non-linear covariates and other potential sources of random errors in the data.

The response variable $y_i$ takes value from the set {0,1} such that $y_i = 1$ if girl $i$ aged 0–14 years has undergone FGM/C and 0, otherwise for $i = 1, \ldots, n$, where $n = 17529$ for the 2016 data and $n = 88319$ for the pooled dataset. This implies that the $y$'s are realizations from Bernoulli trials or $y \sim Bernoulli(p)$, with the probability mass function (pmf) $f(y; p) = p^y (1 - p)^{1-y}$ with $E[Y|p] = p$ and $Var(Y|p) = p(1 - p)$, where $p$ is the probability of success. In other words, $p$ is the probability that a randomly selected Nigerian girl aged 0–14 years has been cut.

Our models follow a class of structured additive regression (STAR; [35–37]) models such that the response $y$ depends on a set of covariates through a linear predictor $\eta_i$ linked to a function of its mean with a link function $h(\mu_i)$ given in Eq (1).

$$h(\mu_i) = \eta_i = f_1(x_{i1}) + \cdots + f_c(x_{ic}) + f_{spat}(s_i) + z_i'\omega \qquad (1)$$

where $h(\mu_i)$ is a logit link function; $f_1, \ldots, f_c$ are the non-linear (not necessarily smooth) functions of continuous covariates $x_{i1}, \ldots, x_{ic}$ (e.g., mother's age, survey year); $f_{spat}(s_i)$ is the (non-parameteric) function of the the spatial covariate $s_i \in \{1, \ldots, S\}$ corresponding to the consecutively numbered geographical locations for the 36 Nigerian states (*state*) and the FCT, that is, $S = 37$, and which accounts for the unobserved effects of geographical locations; and $z_i$'s are categorical variables (e.g., Gender, Educational level, Wealth index, etc) with the coefficients vector $\omega$.

According to the first law of Geography which states that "Everything else is related to each other but near objects are more similar than those further apart" [38], it makes sense to assume that the geographical locations (states) that are near to each other (neighbouring states) are more similar thus with potentially autocorrelated response and it is no longer appropriate to use statistical models that are only valid when observations are independent. Consequently, we assume that observations that are further apart are independent and do not share common boundaries and characteristics thus spatially heterogenous and uncorrelated. As a result, to

simultaneously account for the inherent spatial autocorrelation between states that are neighbours and the spatial independence between states that are further apart, we decompose, the total spatial effect $f_{spat}$ in (1) into a spatially correlated (structured) $f_{str}$ (.) and an uncorrelated (unstructured) $funstr$ (.) effects as in Eq (2) below.

$$f_{spat}(s_i) = f_{str}(s_i) + f_{unstr}(s_i) \qquad (2)$$

This decomposition allows us to explicitly account for and quantify the effects of spatial autocorrelation among neighbouring states and spatial heterogeneity among states that are not neighbours. For these models, all functions are centred on zero for identifiability.

The logit function assumed in (1) allows us to express our full model in terms of log-odds, $\log(p_i/1 - p_i)$, as

$$logit(p_i) = \log\left(\frac{p_i}{1 - p_i}\right) = \eta_i$$
$$= f_{str}(State_i) + f_{unstr}(State_i) + f(Year) + f(Age) + Residence + Education + \cdots$$
$$+ fgm_{woman} + Survey + \xi (3)$$

where the variables *Residence*, *Education*, and $fgm_{woman}$ are fixed effects for residence (rural-urban), mother's education and a mother's FGM/C status. The term *Survey* is a fixed effect accounting for variations due to differences in the different survey methods with DHS used as the reference survey. The term $\xi$ (= *State ID* × *Survey year*) represents space-time interaction effects to account for any other source of variation that varies simultaneously in space and time.

Variants of the model described in (3) are fitted to the 2016 MICS and the pooled 2003 to 2016 datasets within Bayesian statistical framework. These were implemented in R statistical programming software using the R2BayesX package [39], the R interface of BayesX a popular statistical software for fitting various classes of generalized additive mixed models [39–41]. In BayesX, the non-linear functions $f_j$ are modelled as a linear combination of basis functions as

$$f(x) = \sum_{m=1}^{M} \beta_m B_m(x) \qquad (4)$$

where the basis function $B_m$ is known and the unknown vector of regression coefficients $\boldsymbol{\beta} = (\beta_1,...,\beta_M)'$ are to be estimated such that in matrix form, we have $\boldsymbol{f_j} = \boldsymbol{X_j}\boldsymbol{\beta_j}$, where $\boldsymbol{X_j}$ is an n × M design matrix with elements $X[i, m] = B_m(x_i)$. The fixed effects parameters $\omega$ are given diffuse priors such that, $\pi(\omega_j) \propto constant$. Also, a multiplicative normal prior is assumed for the unknown regression parameters such that

$$\pi\left(\beta_j | \tau_j\right) \propto \left(\frac{1}{\tau_j^2}\right)^{\frac{rank(K_j)}{2}} \exp\left(-\frac{1}{2\tau_j^2} \beta_j' K_j \beta_j\right) \qquad (5)$$

where $K_j$ correspond to the frequentist penalty matrix and $\tau_j$ is a smoothing parameter. We assign Markov random fields (MRF; [40]) priors to the correlated spatial effect $f_{str}$ (s), s = 1,..., S, and exploit the neighbourhood structure of the MRF prior to 'borrow strength' from neighbours with more observations to estimate effects in neighbouring states where observations are sparse. Note that the MRF prior is the spatial extension of random walk models and is

given by

$$f_{str}(s) \mid f_{str}(r) \ , \ r \neq s \sim N\left(\sum_{r \sim s} \frac{f_{str}(r)}{N_s}, \frac{\tau_{str}^2}{N_s}\right) \tag{6}$$

where $N_s$ is the number of states that are contiguous (share boarder) to state $s$, and $r \sim s$ denotes that state $r$ is a neighbour of state $s$. Thus, the (conditional) mean of $f_{str}(s)$ is the average of functions $f_{str}(s)$ of the neighbouring states, where $\tau_{str}$ is a variance parameter. In contrast, we assign a zero-mean independent and identically distributed Gaussian priors to the uncorrelated (unstructured) spatial effect $f_{unstr}(s)$ as

$$f_{unstr}(s) \mid \tau_{unstr} \sim N(0, \tau_{unstr}^2), \tag{7}$$

where $\tau_{unstr}$ is a smoothing parameter. We modelled the interaction term $\xi$ as a smooth function with a random walk of the second order (RW2) prior assuming autocorrelation in time and space. However, we also modelled the interaction term assuming the interaction in space and time are not autocorrelated such that $\xi \sim N(0, \ \tau_\xi^2)$. Inverse gamma distributed hyperpriors are then assigned to both the variance and the smooth parameters such that $\pi(\tau_j^2) \sim IG(a_j, b_j)$, where $j$ is a generic subscript representing for example, $str$, $unstr$, $\xi$ and where $a$ and $b$ are hyperparameters.

To estimate the smooth functions, $f_1, \ldots, f_p$, we used cubic splines which are twice continuously differentiable piecewise cubic polynomials. The spline can be written as a linear combination of B-spline basis functions $B_m(x)$, the Bayesian version of the Penalized–Splines (P–Splines) proposed by Eilers & Marx [41], such that $f(x) = \sum_{m=1}^{l} \beta_m B_m(x)$. In our approach, this corresponds to *2nd* order random walks given by

$$\beta_m = 2\beta_{m-1} - \beta_{m-2} + \mu_m \tag{8}$$

with Gaussian increments $\mu_m \sim N(0, \tau^2)$ which is estimated from data and where the smoothness parameter $\tau$ is also estimated from the data.

To address our questions, we tested the following six (6) nested models specified in Table 2. For these models, models $m_1$ & $m_2$ tested the unadjusted effects of a girl's geographical location to her likelihood of being cut while adjusting for the variations due to different survey methods and the year of survey. On the other hand, with models $m_3$ to $m_6$ we simultaneously tested for the effects of geographical locations, social norms, mother's age and other key covariates on a girl's likelihood of being cut while adjusting for variations due to survey methods differences ($m_4$) including potential interactions in space and time ($m_5$ & $m_6$). Note that for the single 2016 MICS data, these models did not include adjustments for survey differences, survey year

**Table 2. Specifications of the model fitted to the datasets.**

| Model | Specification | Remarks |
|---|---|---|
| $m_1$ | $fgm_i \sim f_{str}(state_i) + f_{unstr}(state_i) + f(year)$ | |
| $m_2$ | $fgm_i \sim f_{str}(state_i) + f_{unstr}(state_i) + f(year) + survey$ | |
| $m_3$ | $fgm_i \sim f_{str}(state_i) + f_{unstr}(state_i) + Ethnicity_i + Gender_i + \cdots + f(Age_i) + f(year)$ | |
| $m_4$ | $fgm_i \sim f_{str}(state_i) + f_{unstr}(state_i) + Ethnicity_i + Gender_i + \cdots + f(Age_i) + f(year)$ $+ survey$ | |
| $m_5$ | $fgm_i \sim f_{str}(state_i) + f_{unstr}(state_i) + Ethnicity_i + Gender + \cdots + f(Age_i) + f(year)$ $+ survey + f(\xi)$ | $\xi$ modelled as a smooth function. |
| $m_6$ | $fgm_i \sim f_{str}(state_i) + f_{unstr}(state_i) + Ethnicity_i + Gender_i + \cdots + f(Age_i) + f(year)$ $+ survey + \xi, with \ \xi \sim N(0, \ \sigma_\xi^2)$ | $\xi$ modelled as random effect. |

and space-time interactions so that only two models representing unadjusted and adjusted (full) models are fitted on the 2016 MICS data.

Posterior samples are drawn from the parameters space $\theta$ = ($\{f\}, f_{unstr}, f_{str}, \eta$) and hyper-parameter space $\vartheta$ = ($\tau_{str}, \tau_{unstr}, \tau_{\xi}$) via Markov chain Monte Carlo (MCMC; [42]) simulation. Specifically, we used Metropolis-Hastings (M-H) updating steps with iteratively weighted least square (IWLS) proposal [43].

For our study, 20,000 samples were simulated from the posterior distributions with the hyperparameters set to $a$ = 1, $b$ = 0.0005, that is, diffuse priors. The choice of hyperprior values for the variance parameters were informed by a rigorous sensitivity analysis. Bayesian inference was then based on the last 16000 samples obtained after discarding the first 4000 as burn-in. The last 16000 samples were further *thinned* so that only every 20th value was included among the values from which the posterior estimates were evaluated. Both the burn-in and thinning are used to minimize autocorrelations within the posterior samples.

In addition, we investigated the appropriateness of the Markov Random Field (MRF) priors through sensitivity analysis by fitting the spatial model using Gaussian Random Field (GRF; [37]) priors. However, we found no evidence of a better fit with the GRF. Besides, the sparseness introduced by the neighbourhood structure of the MRF offered a computational advantage and greatly reduced computational costs thus the models fitted with MRF priors were retained and further assessed.

Finally, we assessed how well the models fit the data and the best fit models were selected based on the Deviance Information Criterion (DIC; [44]). Results are presented as posterior odds ratios (POR), graphs and maps. Maps were produced in R statistical software version 4.0.0, while all boundary files (shapefiles) were freely available and downloaded from DIVA-GIS (www.diva-gis.org/gdata) with permission to publish obtained from Global Administrative Areas (GADM). With respect to handling missing values, the bivariate descriptive analysis using Stata and the Bayesian geo-additive multilevel regression models using R employed list-wise deletion algorithm for observations with missing values. Note that women included in the analysis were only those with living daughters aged 0–14 years old and those who were able to provide information on their daughters' FGM/C statuses.

## Results

### Descriptive statistics

Results from the bivariate analysis of the baseline characteristics of the women and their daughters are presented in Table 3. These results suggest that FGM/C prevalence among 0-14-year-old girls in Nigeria rose to its peak in 2008 with respect to most of the baseline characteristics ($p < 0.0001$) but showed a general decline in 2016. For example, in 2008, FGM/C prevalence was highest among daughters of older women (aged 30–49). However, in 2016 prevalence declined by 28.1% among daughters of women aged 45–49 years. In contrast, between 2008 and 2016, FGM/C prevalence rose by 14.9% among daughters of younger women aged 15–19 years. Other individual-level characteristics which showed significant association with a girl's FGM/C status are her ages at the time of the survey and at cutting ($p < 0.0001$). Most of the girls who were found to have undergone FGM/C were aged between 5–14 years as at the time of the surveys, while the vast majority of the girls (up to 97.7% in 2013) were cut before attaining their 5th birthday with up to 81% of the cutting carried out by Traditional circumcisers in 2016. FGM/C prevalence was highest among daughters of women who alongside their husbands/partners had no formal education. Also, across household wealth indices we found overall decline in FGM/C prevalence between 2008 and 2016, except for the lowest and second indices, which increased by 16.2% and 0.4%, respectively. High

**Table 3. Characteristics surrounding female genital mutilation/cutting in Nigerian girls aged 0–14 from 2003–2017.**

| Demographic characteristic | 2003 DHS | | 2007 MICS | | 2008 DHS | | 2011 MICS | | 2013 DHS | | 2016–17 MICS | |
|---|---|---|---|---|---|---|---|---|---|---|---|---|
| | N = 3281, FGM = 17.3% | | N = 7768, FGM = 22.4% | | N = 17691, FGM = 30.0% | | N = 16874, FGM = 19.2% | | N = 25176, FGM = 24.4% | | N = 17529, FGM = 25.3% | |
| | Number (%) | P-value | Number (%) | P-value | Number (%) | P-value | Number (%) | P-value | Number (%) | P-value | Number (%) | P-value |
| **Girl's age** | | < 0.0001 | | | | < 0.0001 | | | | <0.0001 | | |
| 0–4 | 1324 (12.6) | | - | | 7165 (26.8) | | | | 9839 (23.3) | | - | |
| 5–9 | 1037 (17.3) | | - | | 5882 (31.2) | | - | | 8737 (25.2) | | - | |
| 10–14 | 920 (23.9) | | - | | 4645 (33.5) | | - | | 6599 (25.1) | | - | |
| **Mother's age** | | 0.0028 | | <0.0001 | | <0.0001 | | 0.0069 | | 0.1373 | | <0.0001 |
| 15–19 | 64 (1.3) | | 136 (17.8) | | 253 (29) | | 194 (27) | | 425 (33.5) | | 257 (43.9) | |
| 20–24 | 284 (14.6) | | 582 (14.2) | | 1525 (26.5) | | 1296 (18.9) | | 2179 (26.3) | | 1476 (29.3) | |
| 25–29 | 736 (11.8) | | 1514 (15.4) | | 3800 (26.9) | | 3435 (16.3) | | 5243 (24.4) | | 3210 (27.4) | |
| 30–34 | 747 (10.9) | | 1729 (19.5) | | 4251 (26.6) | | 4404 (19.5) | | 5824 (23.6) | | 4681 (22.6) | |
| 35–39 | 729 (18.6) | | 1585 (23.4) | | 3817 (30.4) | | 3552 (18) | | 5722 (23.6) | | 4811 (18.6) | |
| 40–44 | 471 (28.5) | | 1219 (29.3) | | 2613 (36.6) | | 2609 (19.9) | | 3499 (24.2) | | 4677 (14.2) | |
| 45–49 | 250 (34.4) | | 1003 (33.5) | | 1432 (39.6) | | 1385 (26.8) | | 2283 (25.5) | | 3473 (11.5) | |
| **Mother's marital status** | | 0.1616 | | 0.3785 | | 0.01 | | 0.2398 | | 0.0012 | | <0.0001 |
| Never married | 49 (10.2) | | 167 (18.7) | | 205 (14.3) | | 185 (20) | | 223 (11.9) | | 197 (6.1) | |
| Currently married/in union | 3079 (17.7) | | 7.04 (22.3) | | 16642 (30.2) | | 15809 (19.5) | | 23800 (24.7) | | 16398 (25.9) | |
| Formerly married | 153 (10.4) | | 559 (25) | | 844 (30.9) | | 879 (13.5) | | 1153 (21.8) | | 916 (18.9) | |
| **Mother's age difference with husband/ partner (currently married women only)** | | 0.2873 | | 0.9739 | | 0.0049 | | 0.2616 | | 0.0018 | | <0.0001 |
| Wife is older | 36 (8.0) | | 115 (20.3) | | 214 (18.6) | | 324 (20.8) | | 200 (19.9) | | 565 (48.7) | |
| Wife is same age | 12 (4.8) | | 47 (23.9) | | 228 (32.6) | | 188 (21.8) | | 288 (15.4) | | 244 (24.8) | |
| Wife is 1–4 years younger | 410 (22.9) | | 1177 (22.3) | | 2726 (29.2) | | 2903 (17.7) | | 3509 (19.9) | | 2528 (19.1) | |
| Wife is 5–9 years younger | 1076 (16) | | 2234 (23) | | 5415 (27.2) | | 4732 (16.2) | | 8022 (24.4) | | 4951 (21.5) | |
| Wife is 10+ years younger | 1749 (17) | | 4195 (22.2) | | 9107 (32.2) | | 8726 (21.2) | | 13156 (26) | | 9243 (27.7) | |
| **Mother's type of union** | | 0.0094 | | - | | 0.0001 | | <0.0001 | | <0.0001 | | <0.0001 |
| Monogamous | 2033 (20.0) | | - | | 11624 (27.8) | | 11114 (17.5) | | 15546 (22.5) | | 11086 (20.6) | |
| Polygamous | 1032 (13.3) | | - | | 4915 (35.7) | | 4544 (24.8) | | 8082 (29.1) | | 5291 (37.2) | |
| **Residence** | | 0.1894 | | 0.0616 | | 0.1089 | | 0.0019 | | 0.0001 | | 0.0008 |

*(Continued)*

**Table 3.** (Continued)

| Demographic characteristic | 2003 DHS | | 2007 MICS | | 2008 DHS | | 2011 MICS | | 2013 DHS | | 2016–17 MICS | |
|---|---|---|---|---|---|---|---|---|---|---|---|---|
| | N = 3281, FGM = 17.3% | | N = 7768, FGM = 22.4% | | N = 17691, FGM = 30.0% | | N = 16874, FGM = 19.2% | | N = 25176, FGM = 24.4% | | N = 17529, FGM = 25.3% | |
| | Number (%) | P-value | Number (%) | P-value | Number (%) | P-value | Number (%) | P-value | Number (%) | P-value | Number (%) | P-value |
| Urban | 1354 (20.3) | | 3299 (24.4) | | 6957 (27) | | 6866 (15.4) | | 10577 (20.7) | | 7254 (20.5) | |
| Rural | 1927 (15.1) | | 4469 (21) | | 10734 (32) | | 10008 (21.9) | | 14598 (27.1) | | 10276 (28.8) | |
| **Zone** | | <0.0001 | | <0.0001 | | <0.0001 | | <0.0001 | | <0.0001 | | <0.0001 |
| North Central | 389 (17.2) | | 806 (16.6) | | 1368 (20) | | 1756 (11.5) | | 1630 (12.8) | | 2323 (16.1) | |
| North East | 566 (0.6) | | 664 (2.3) | | 1869 (6.3) | | 1654 (4.7) | | 3562 (8.3) | | 2761 (1.4) | |
| North West | 556 (2.6) | | 801 (8.7) | | 3856 (46.3) | | 2937 (37.3) | | 10153 (35.5) | | 5329 (56) | |
| South East | 473 (28.8) | | 982 (24.1) | | 2725 (29.9) | | 2697 (14.8) | | 2747 (23.5) | | 1476 (12.7) | |
| South South | 791 (18.0) | | 1753 (17.9) | | 3274 (18.5) | | 3389 (11.6) | | 2882 (8.1) | | 2311 (6.1) | |
| South West | 506 (40.1) | | 2762 (35.2) | | 4600 (37.2) | | 4442 (24.2) | | 4203 (27.8) | | 3329 (21.6) | |
| **Mother's education** | | 0.0562 | | <0.0001 | | <0.0001 | | 0.0095 | | <0.0001 | | <0.0001 |
| No education | 1106 (13.5) | | 92 (16.1) | | 5895 (35.3) | | 610 (31.2) | | 11638 (30.2) | | 3002 (44.2) | |
| Primary | 961 (20.5) | | 2301 (28.4) | | 4838 (31.8) | | 4263 (19) | | 5396 (23.4) | | 3378 (23.6) | |
| Secondary | 977 (20.3) | | 2498 (23) | | 5436 (26.5) | | 5976 (17.9) | | 6349 (18.5) | | 5576 (17.2) | |
| Higher | 237 (9.2) | | 681 (13.7) | | 1522 (16.5) | | 1852 (7) | | 1793 (10.6) | | 1909 (9.8) | |
| **Husband's/partner's education** | | 0.0112 | | - | | <0.0001 | | - | | <0.0001 | | - |
| No education | 816 (14.1) | | - | | 4695 (34.9) | | - | | 9542 (30.7) | | - | |
| Primary | 850 (24.6) | | - | | 4403 (32.7) | | - | | 4929 (25.5) | | - | |
| Secondary | 912 (17.9) | | - | | 5636 (29.5) | | - | | 6898 (20.3) | | - | |
| Higher | 600 (11.2) | | - | | 2498 (18.3) | | - | | 3370 (14.6) | | - | |
| **Mother's religion** | | 0.002 | | 0.7127 | | <0.0001 | | <0.0001 | | <0.0001 | | - |
| Christian | 1881 (21.2) | | 4907 (22.6) | | 9719 (22.8) | | 9792 (14.1) | | 9618 (14.8) | | - | |
| Muslim | 1348 (11.7) | | 2717 (22) | | 7530 (39.3) | | 6830 (26.3) | | 15212 (30.6) | | - | |
| Other | 52 (21.7) | | 144 (26.3) | | 385 (32.6) | | 241 (27.3) | | 227 (20.2) | | - | |
| **Mother's ethnicity** | | <0.0001 | | <0.0001 | | <0.0001 | | - | | <0.0001 | | <0.0001 |
| Fulani | 82 (0.8) | | 170 (5.1) | | 657 (27.4) | | - | | 1654 (28.8) | | - | |
| Hausa | 639 (2.9) | | 768 (7.6) | | 3682 (46.9) | | - | | 9581 (33.7) | | 7785 (38.6) | |

*(Continued)*

**Table 3.** (Continued)

| Demographic characteristic | 2003 DHS | | 2007 MICS | | 2008 DHS | | 2011 MICS | | 2013 DHS | | 2016–17 MICS | |
|---|---|---|---|---|---|---|---|---|---|---|---|---|
| | N = 3281, FGM = 17.3% | | N = 7768, FGM = 22.4% | | N = 17691, FGM = 30.0% | | N = 16874, FGM = 19.2% | | N = 25176, FGM = 24.4% | | N = 17529, FGM = 25.3% | |
| | Number (%) | P-value | Number (%) | P-value | Number (%) | P-value | Number (%) | P-value | Number (%) | P-value | Number (%) | P-value |
| Igbo | 649 (27.9) | | 1081 (26.3) | | 3428 (29.6) | | - | | 3387 (21) | | 2153 (11.3) | |
| Kanuri | 110 (0) | | 69 (0) | | 543 (3.5) | | - | | 505 (5.4) | | | |
| Tiv | 53 (0) | | 112 (0) | | 316 (2.2) | | - | | 233 (1.3) | | | |
| Yoruba | 484 (45.9) | | 1582 (51.1) | | 3963 (42.1) | | - | | 3823 (32.4) | | 2984 (27.4) | |
| Other | 1264 (11.3) | | 3986 (14.6) | | 5016 (13.8) | | - | | 5993 (7.8) | | 4608 (8.3) | |
| **Woman from mixed ethnicity household (husband/partner from a different ethnic group; currently married women only)** | | <0.0001 | | - | | 0.0026 | | - | | 0.0102 | | - |
| Yes | 0 (0) | | - | | 597 (20.9) | | - | | 993 (16.7) | | - | |
| No | 732 (14.5) | | - | | 5490 (33.8) | | - | | 6559 (25.4) | | - | |
| **Wealth Quintile** | | 0.2364 | | <0.0001 | | 0.0243 | | <0.0001 | | <0.0001 | | <0.0001 |
| Lowest | 494 (16.4) | | 764 (12.6) | | 2425 (26.8) | | 2099 (24.7) | | 5600 (29.8) | | 2209 (43) | |
| Second | 512 (17.3) | | 1075 (21.7) | | 3064 (37.3) | | 2843 (23.1) | | 5030 (32.5) | | 2633 (37.7) | |
| Middle | 555 (17.2) | | 1408 (22.6) | | 3497 (30.5) | | 3574 (20.8) | | 4412 (23.7) | | 3737 (25.7) | |
| Fourth | 781 (12.5) | | 2100 (29.5) | | 4149 (30.8) | | 4238 (21.4) | | 4934 (20.7) | | 4414 (20.1) | |
| Highest | 939 (21.7) | | 2422 (19.6) | | 4556 (25.8) | | 4120 (10.2) | | 5199 (15) | | 4537 (14.5) | |
| **Age at Cutting for girl** | | <0.0001 | | <0.0001 | | <0.0001 | | <0.0001 | | <0.0001 | | <0.0001 |
| 0–4 | 504 (90.7) | | 1219 (70) | | 5041 (94.9) | | 2906 (89.6) | | 6009 (97.7) | | - | |
| 5–9 | 24 (4.3) | | 275 (15.8) | | 127 (2.4) | | 68 (2.1) | | 68 (1.1) | | - | |
| 10–14 | 12 (2.2) | | 96 (5.5) | | 37 (0.7) | | 10 (0.3) | | 12 (0.2) | | - | |
| 15+ | - | | - | | - | | - | | - | | - | |
| Missing/Don't know | 16 (2.8) | | 153 (8.8) | | 106 (2) | | 259 (8) | | 62 (1) | | - | |
| **Person who performed cutting For girl** | | | | <0.0001 | | <0.0001 | | <0.0001 | | <0.0001 | | <0.0001 |
| Doctor | - | | 1059 (60.8) | | 142 (2.7) | | 117 (3.6) | | 86 (0.9) | | 102 (2.3) | |
| Nurse/Midwife/Other health worker | - | | 85 (4.9) | | 888 (16.9) | | 795 (24.5) | | 1051 (11) | | 444 (10) | |
| Traditional circumciser | - | | 153 (8.8) | | 3793 (72.2) | | 2131 (65.7) | | 8029 (84) | | 3599 (81) | |
| Other traditional practitioners/ TBA | - | | 428 (24.6) | | 420 (8) | | 130 (4) | | 249 (2.6) | | 284 (6.4) | |
| Don't know/Missing | - | | 16 (0.9) | | 16 (0.3) | | 71 (2.2) | | 143 (1.5) | | 13 (0.3) | |
| **Type of FGM/C among girls** | | <0.0001 | | <0.0001 | | <0.0001 | | <0.0001 | | <0.0001 | | <0.0001 |

*(Continued)*

**Table 3.** (Continued)

| Demographic characteristic | 2003 DHS | | 2007 MICS | | 2008 DHS | | 2011 MICS | | 2013 DHS | | 2016–17 MICS | |
|---|---|---|---|---|---|---|---|---|---|---|---|---|
| | N = 3281, FGM = 17.3% | | N = 7768, FGM = 22.4% | | N = 17691, FGM = 30.0% | | N = 16874, FGM = 19.2% | | N = 25176, FGM = 24.4% | | N = 17529, FGM = 25.3% | |
| | Number (%) | P-value | Number (%) | P-value | Number (%) | P-value | Number (%) | P-value | Number (%) | P-value | Number (%) | P-value |
| Sewn Closed | 23 (4.1) | | 188 (10.8) | | 418 (7.9) | | 185 (5.7) | | 160 (2.6) | | 235 (5.3) | |
| Not sewn closed | 406 (73) | | 1092 (62.7) | | 3624 (68.5) | | 2740 (84.5) | | 5689 (92.5) | | 4088 (92) | |
| Don't know/missing | 127 (22.9) | | 461 (26.5) | | 1248 (23.6) | | 318 (9.8) | | 301 (4.9) | | 120 (2.7) | |
| **Number of cut women (girls)** | 1445 {556} | | 6375 {1741} | | 9890 {5290} | | 8321 {3243} | | 9651 {6150} | | 6312 {4443} | |
| **Mother's support for FGM/C continuation** | | <0.0001 | | - | | <0.0001 | | <0.0001 | | <0.0001 | | <0.0001 |
| Continued | 613 (51) | | - | | 4048 (76.1) | | 3788 (58.8) | | 6086 (61.5) | | 5308 (58.9) | |
| Discontinued | 2287 (8.8) | | - | | 11002 (10.7) | | 11133 (6.6) | | 15816 (10.5) | | 14862 (5.3) | |
| Depends/ Don't know | 364 (14.3) | | - | | 2594 (39.8) | | 1928 (14.3) | | 3038 (23.6) | | 2412 (21.9) | |
| **Mother's beliefs about FGM/C** | | <0.0001 | | - | | <0.0001 | | - | | <0.0001 | | - |
| FGM/C is required by religion | 421 (41.6) | | - | | 2944 (60.1) | | - | | 3888 (51.4) | | - | |
| FGM/C is not required by religion | 2466 (13.1) | | - | | 12144 (20.8) | | - | | 16855 (17.9) | | - | |
| FGM/C prevents girl's premarital sex | 371 (29.3) | | - | | 1935 (60.8) | | - | | - | | - | |
| FGM/C does not prevent girl's premarital sex | 1822 (19.6) | | - | | 15670 (26.2) | | - | | - | | - | |
| **Final Say in making decision on large household purchases** | | 0.0022 | | | | <0.0001 | | - | | - | | - |
| Mother only | 433 (25.6) | | - | | 1023 (25.3) | | - | | - | | - | |
| Jointly with partner/someone else | 605 (21) | | - | | 6113 (25.1) | | - | | - | | - | |
| Partner/Someone else only | 2222 (14.3) | | - | | 9437 (34) | | - | | - | | - | |
| **Final Say in making decision on mother's own health** | | <0.0001 | | | | 0.0001 | | - | | - | | - |
| Mother only | 720 (29.6) | | - | | 2005 (32) | | - | | - | | - | |
| Jointly with partner/someone else | 467 (19.1) | | - | | 6431 (24.2) | | - | | - | | - | |
| Partner/Someone else only | 2084 (12.3) | | - | | 8137 (34.5) | | - | | - | | - | |
| **Mother employed in the last 7 days** | | 0.0068 | | - | | 0.7200 | | - | | 0.2273 | | - |
| Yes | 679 (8.3) | | - | | 3659 (30.7) | | - | | 6146 (22.6) | | - | |
| No | 2599 (19.6) | | - | | 13924 (29.8) | | - | | 18958 (25) | | - | |
| **Mother's Occupation** | | 0.0036 | | - | | 0.0001 | | - | | <0.0001 | | - |

*(Continued)*

**Table 3.** (Continued)

| Demographic characteristic | 2003 DHS | | 2007 MICS | | 2008 DHS | | 2011 MICS | | 2013 DHS | | 2016–17 MICS | |
|---|---|---|---|---|---|---|---|---|---|---|---|---|
| | N = 3281, FGM = 17.3% | | N = 7768, FGM = 22.4% | | N = 17691, FGM = 30.0% | | N = 16874, FGM = 19.2% | | N = 25176, FGM = 24.4% | | N = 17529, FGM = 25.3% | |
| | Number (%) | P-value | Number (%) | P-value | Number (%) | P-value | Number (%) | P-value | Number (%) | P-value | Number (%) | P-value |
| Formal | 335 (10.9) | | - | | 1147 (16.5) | | - | | 1262 (12.6) | | - | |
| Informal | 2321 (20.5) | | - | | 13349 (31.9) | | - | | 17813 (25.9) | | - | |
| Not working | 626 (8.5) | | - | | 3119 (27.4) | | - | | 5845 (23.3) | | - | |
| **Mother's employment type** | | 0.0375 | | - | | 0.0001 | | - | | 0.2276 | | - |
| All year | 2001 (19.5) | | - | | 11140 (32.8) | | - | | 16363 (25.4) | | - | |
| Seasonal/part of the year/once in a while | 645 (18.7) | | - | | 3365 (23.4) | | - | | 2927 (21.2) | | - | |
| **Woman works for cash/cash and kind** | | 0.0111 | | - | | 0.0002 | | - | | 0.0008 | | - |
| Yes | 331 (25.7) | | - | | 2488 (22.1) | | - | | 1143 (14.2) | | - | |
| No | 2320 (18.4) | | - | | 12039 (32.3) | | - | | 18127 (25.5) | | - | |
| **Mother's income (currently married women only)** | - | | | | | - | | | | <0.0001 | | |
| Less money than her husband's/partner's | - | | - | | 9208 (30.5) | | - | | 14795 (26.8) | | - | |
| More money than her husband's/partner's | - | | - | | 482 (25.8) | | - | | 715 (16.3) | | - | |
| About the same | - | | - | | 600 (33.9) | | - | | 896 (23.2) | | - | |
| Husband/Partner doesn't bring in any money | - | | - | | 99 (23.3) | | - | | 113 (10) | | - | |
| Don't know | - | | - | | 912 (56.5) | | - | | 507 (16.9) | | - | |
| **Who usually decides mother's cash expenditure** | | 0.4073 | | - | | 0.0001 | | - | | <0.0001 | | - |
| Mother only | 1567 (19.7) | | - | | 7478 (35.5) | | - | | 12594 (27.9) | | - | |
| Jointly with partner/someone else | 513 (16) | | - | | 2371 (25.5) | | - | | 3171 (18.6) | | - | |
| Partner/Someone else only | 241 (15.4) | | - | | 1450 (28.7) | | - | | 1260 (22.7) | | - | |
| **Who usually decides husband's/partner's cash expenditure** | - | | - | | | <0.0001 | | - | | <0.0001 | | - |
| Mother only | - | | - | | 972 (17.1) | | - | | 862 (24.4) | | - | |
| Jointly with husband/partner | - | | - | | 4313 (22.2) | | - | | 5354 (18.5) | | - | |
| Husband/partner only | - | | - | | 11128 (34.6) | | - | | 17270 (26.9) | | - | |
| Husband/Partner has no earnings | - | | - | | 97 (9.5) | | - | | 151 (8) | | - | |
| **MOBILITY** | | | | | | | | | | | | |

(*Continued*)

**Table 3.** (Continued)

| Demographic characteristic | 2003 DHS | | 2007 MICS | | 2008 DHS | | 2011 MICS | | 2013 DHS | | 2016–17 MICS | |
|---|---|---|---|---|---|---|---|---|---|---|---|---|
| | N = 3281, FGM = 17.3% | | N = 7768, FGM = 22.4% | | N = 17691, FGM = 30.0% | | N = 16874, FGM = 19.2% | | N = 25176, FGM = 24.4% | | N = 17529, FGM = 25.3% | |
| | Number (%) | P-value | Number (%) | P-value | Number (%) | P-value | Number (%) | P-value | Number (%) | P-value | Number (%) | P-value |
| **Number of years mother lived continuously in her current location** | | 0.0472 | | - | | 0.2678 | | - | | - | | - |
| 0 years | 92 (4.2) | | - | | 717 (26.1) | | - | | - | | - | |
| 1–10 years | 1135 (16.5) | | - | | 7925 (27.9) | | - | | - | | - | |
| 11–20 years | 631 (22.9) | | - | | 3353 (31) | | - | | - | | - | |
| 21 or more years | 216 (17.3) | | - | | 807 (27.1) | | - | | - | | - | |
| **Mother's number of trips away from the community in the last 12 months** | - | | - | - | | <0.0001 | | - | | 0.1101 | | - |
| 0 | - | | - | | 9687 (32.7) | | - | | 13990 (25.1) | | - | |
| 1–25 | - | | - | | 7881 (26.5) | | - | | 10981 (23.7) | | - | |
| 26–50 | - | | - | | 28 (29.8) | | - | | 116 (20) | | - | |
| 51 or more | - | | - | | 1 (0) | | - | | 88 (14.9) | | - | |
| **MASS MEDIA EXPOSURE** | | | | | | | | | | | | |
| **Frequency of reading newspaper/ magazine** | | 0.8311 | | - | | <0.0001 | | - | | <0.0001 | | - |
| Not at all | 2454 (16.8) | | - | | 13683 (32.3) | | - | | 21115 (26.2) | | - | |
| Less than once a week | 376 (17.4) | | - | | 2027 (22.5) | | - | | 2278 (16.7) | | - | |
| At least once a week | 440 (20.2) | | - | | 1877 (21.4) | | - | | 1645 (12.5) | | - | |
| **Frequency of listening to radio** | | 0.3202 | | - | | 0.0011 | | - | | 0.0001 | | - |
| Not at all | 663 (15.6) | | - | | 4273 (24.9) | | - | | 8666 (25.2) | | - | |
| Less than once a week | 555 (22.9) | | - | | 2837 (31.4) | | - | | 6238 (27.9) | | - | |
| At least once a week | 2053 (16.4) | | - | | 10513 (31.8) | | - | | 10189 (21.8) | | - | |
| **Frequency of watching tv** | | 0.4427 | | - | | 0.0245 | | - | | <0.0001 | | - |
| Not at all | 1601 (16) | | - | | 7858 (33.4) | | - | | 12543 (27.7) | | - | |
| Less than once a week | 365 (14.7) | | - | | 2211 (27.7) | | - | | 4569 (28) | | - | |
| At least once a week | 1314 (19.5) | | - | | 7542 (27.3) | | - | | 7964 (17.2) | | - | |

Note. Blank space indicates variable is missing. Values (in bold) along the characteristics are p-values

*In the 2003 DHS, 2007 MICS, and 2008 DHS, FGM/C questions were asked about the most recently cut daughters of any age; for this analysis, sample size is limited to most recently cut girls aged 0–14. In the 2011 MICS, 2013 DHS and 2016–17 MICS the FGM/C questions were asked for all daughters aged 0–14 years

** MICS 2007 & 2016–17 asked religion and ethnicity of the head of the household

FGM/C prevalence was associated with daughters of women who had undergone FGM/C, supported the continuation of the practice, and women who believed that FGM/C was required by religion and prevents premarital sex. Girls who lived in polygamous households and girls whose mothers are the key decision makers in their households in terms of the woman's health and spending, had highest prevalence of FGM/C. Before 2008, FGM/C was highest in South western and South eastern states in Nigeria, but there has been a shift to the North with the North-western zone accounting for more than half of the cut girls in 2016. Similarly, prior to 2008, prevalence of FGM/C was highest among girls who lived in the urban areas and among Yoruba and Igbo girls, but this has changed with the vast majority of the cut girls being rural dwellers and from Hausa ethnic group since 2008. Finally, the results show that the prevalence of FGM/C was highest among daughters of women who never read newspapers nor watched television, however, the nature of the association between how often a woman listens to radio and her daughter's FGM/C status was not clear.

## Spatial distribution of FGM/C across Nigerian states and regions

The spatial distribution of FGM/C prevalence (crude prevalence) among 0–14 years old girls in Nigerian is presented in Fig 3. Unadjusted estimates of FGM/C prevalence varied across geographical locations (states and the FCT) in Nigeria with the highest prevalence found among the South western states of Ekiti, Ondo, Osun and Oyo, and North-central state of Kwara in 2003 up to 2007. Surprisingly, in 2008, highest prevalence was found in the North

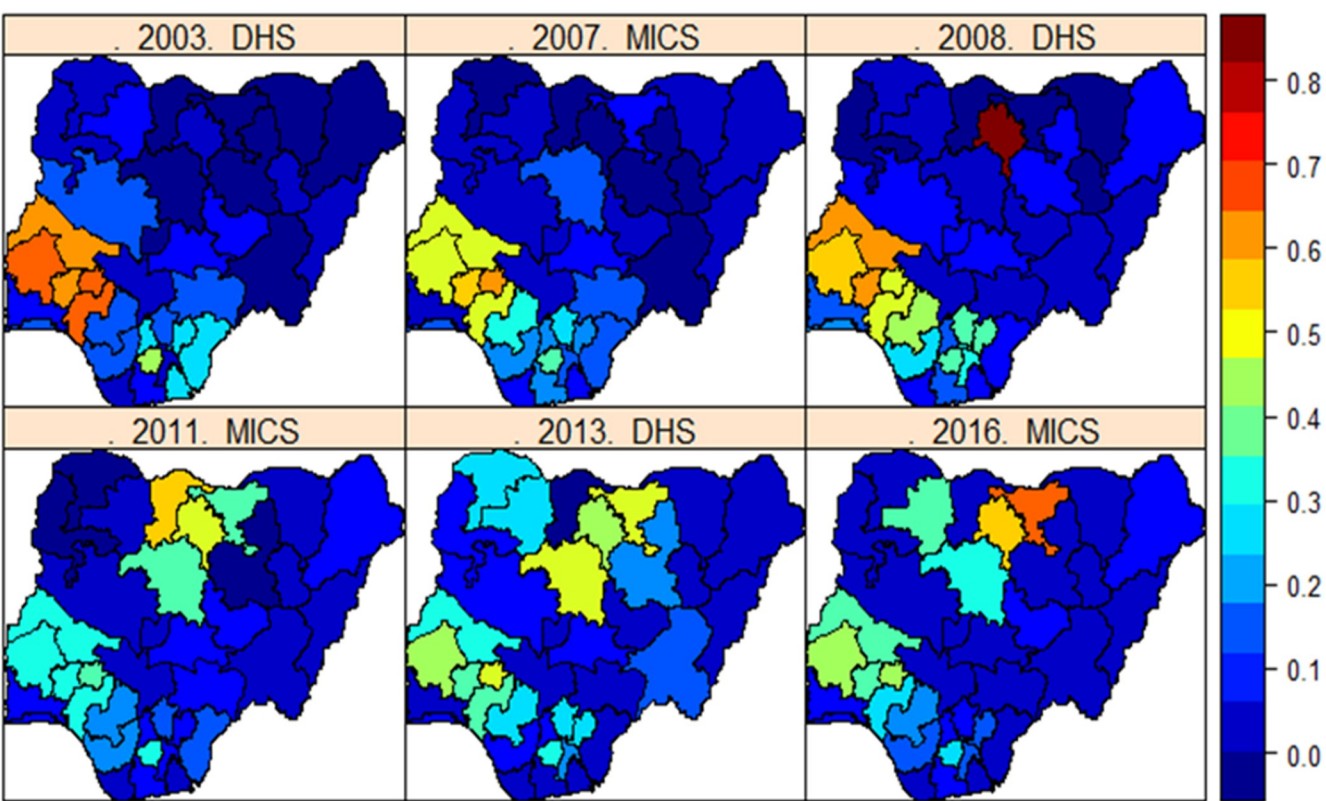

**Fig 3. Crude estimates of the spatial distribution of FGM/C prevalence among 0-14-year old girls across the Nigerian 36 states and the FCT.** Dark blue to dark red correspond to lowest to highest prevalent states. Shapefile republished from DIVA-GIS database (https://www.diva-gis.org/) under a CC BY license, with permission from Global Administrative Areas (GADM), original copyright 2018.

western state of Kano with prevalence in South-western states still high at around 60%. Another rather surprising outcome is the unexpected high FGM/C prevalence found in Katsina state in 2011 and which radically declined to almost 0% afterwards. It is not clear, why such unexpected variations in prevalence of FGM/C existed in Kano and Katsina states. However, it could be that prior to 2008, there were no meaningful survey coverage in the North and data collection in these states has improved ever since. How likely is this? Another potential reason could be due to the different survey methods used by the DHS and MICS and the different definitions of the *fgm* variables used by the surveys before being standardised in 2010 [30]. From 2008 to 2016, FGM/C prevalence remained high among the North-western states with Jigawa state having the highest prevalence in 2016, while a significant decline was found in the Southern states in the same period.

### Bayesian hierarchical spatial models

Using Moran's I spatial clustering test, we found the existence of local spatial clusters in Kano, Kaduna and Jigawa states with Moran's I test statistics of 0.16, p-value = 0.01. In addition, Moran's I Monte Carlo simulations with 1000 samples returned a significant Moran's I test statistics of 0.16 with p-value = 0.02 which further confirmed that indeed FGM/C exhibited positive spatial clustering among the North western states.

In what follows, we present the results obtained from the Bayesian hierarchical models fitted on the various datasets. First, we present the results based on the 2016 MICS data in order examine a 'snapshot' of changes in a girl's likelihood of undergoing FGM/C. Afterwards, we present results based on the pooled 2003–2016 dataset and assess changes over time.

**2016 MICS dataset.** *Posterior Odds Ratio (POR).* Table 4 presents results based on the Bayesian hierarchical spatial models fitted to the 2016 MICS data. Estimates of the DIC for the unadjusted model $m_1^*$ ($m_1$ without year component) and the adjusted model $m_3^*$ ($m_3$ without year component) are 12838.40 and 6236.40, respectively. This indicates that the full model which has the smaller DIC value provided a better fit to the data. However, graphical results from both models are presented for comparison purposes where necessary. Therefore, in Table 4, only the results based on $m_3^*$ are presented.

Daughters of women who did not have up to higher education had significantly higher likelihood of experiencing FGM/C. A girl who lived in the North west is about 6 times more likely to be cut than a girl who lived in the North central states of Nigeria. Also, likelihood of cutting was high among girls whose mothers were formerly married, whose mothers have undergone FGM/C and almost 14 times higher among girls whose mothers supported the continuation of FGM/C in Nigeria. Lower likelihood of FGM/C is associated with girls from other ethnic minorities, and there is no significant difference in likelihood of FGM/C found among girls who lived in households classified under different wealth quintiles suggesting that FGM/C is practised in Nigeria regardless of whether household is rich or poor. In addition, although the likelihood of FGM/C is lower among urban girls, we note that this is not significant suggesting that as of 2016, type of place of residence is not key to determining the likelihood of cutting a Nigerian girl aged 0–14 years.

*Effects of geographical location.* In Fig 4, we show the posterior means (top panel) of the effects of geographical location on a girl's likelihood of being cut, and the corresponding maps testing the statistical significance of the estimates based on 95% credible interval (bottom panel) for the unadjusted model ($m_1^*$;left) and the adjusted ($m_3^*$;right) models fitted to the 2016 MICS dataset. For these maps, dark blue to dark red correspond to lowest risk to highest risk states. While black, white and grey correspond to significantly high-risk states, significantly low risk states, and the states in which the effects of geographical location were not statistically

**Table 4. Posterior odds ratio estimates from the Bayesian geo-additive regression model (adjusted) fitted to the 2016 MICS data.**

| Variable | | POR | CI | |
|---|---|---|---|---|
| Name | Remarks/level | | 2.5% | 97.5% |
| *Education* | Higher (ref) | 1.00 | - | - |
| | None | 1.623 | 1.131 | 2.353 |
| | Primary | 1.352 | 1.352 | 1.352 |
| | Secondary | 1.355 | 1.045 | 1.719 |
| *Women support for fgm/c continuation* | Stopped (ref) | 1.00 | | |
| | Continued | 13.944 | 11.968 | 16.291 |
| | Depends/don't know | 3.111 | 2.463 | 3.855 |
| *Wealth index* | Middle (ref) | 1.00 | | |
| | Highest | 1.045 | 0.817 | 1.332 |
| | Higher | 1.056 | 0.852 | 1.279 |
| | Second | 1.209 | 0.956 | 1.574 |
| | Lowest | 1.181 | 0.913 | 1.539 |
| *Ethnicity* | Hausa (ref) | 1.00 | | |
| | Igbo | 0.78 | 0.488 | 1.236 |
| | Yoruba | 1.059 | 0.687 | 1.629 |
| | Others | 0.39 | 0.278 | 0.554 |
| *Region* | North central | 1.00 | | |
| | North east | 0.258 | 0.037 | 2.025 |
| | North west | 6.009 | 1.048 | 38.133 |
| | South east | 1.263 | 0.158 | 6.595 |
| | South-south | 0.955 | 0.12 | 5.607 |
| | South west | 1.43 | 0.295 | 10.274 |
| *$fgm_{woman}$* | Not cut (ref) | 1.00 | | |
| | Cut | 11.637 | 9.838 | 14.165 |
| *Place of residence* | Rural (ref) | 1.00 | | |
| | Urban | 0.954 | 0.797 | 1.137 |
| *Marital status* | Currently married (ref) | 1.00 | | |
| | Formerly married | 1.422 | 1.05 | 1.892 |
| | Never married | 0.671 | 0.293 | 1.387 |

POR- Posterior Odds Ration; CI- Credible Interval; SD- Standard deviation.

significant. The unadjusted maps show that significantly higher likelihood of cutting existed among girls who lived in the North western states of Zamfara, Kaduna, Kano and Jigawa; South western states of Ekiti, Oyo, Osun, and Ondo; North central state of Kwara and Plateau; South southern state of Edo and South eastern state of Imo. However, after taking account of the effects of other key covariates considered in the model (Table 4), only the effects of geographical location in Jigawa and Plateau states remained significantly high. This suggests that the key factors driving FGM/C among 0–14 years old in Nigeria are mainly state-specific or due to sharing boundary with high prevalent states (e.g., Jigawa and Kano).

**Pooled 2003 to 2016 dataset.** In this Section, we present the results from the Bayesian hierarchical models fitted to the pooled dataset. Unlike the 2016 MICS data, the pooled data allows us to assess changes over time and space and better estimate prevalence.

*DIC.* Fig 5 shows estimates of the Deviance Information Criterion (DIC) of the six (6) nested models specified in Table 2 and fitted to the pooled 2003 to 2016 dataset. Based on the DIC, Fig 5A shows that the unadjusted models fit the data equally well regardless of whether

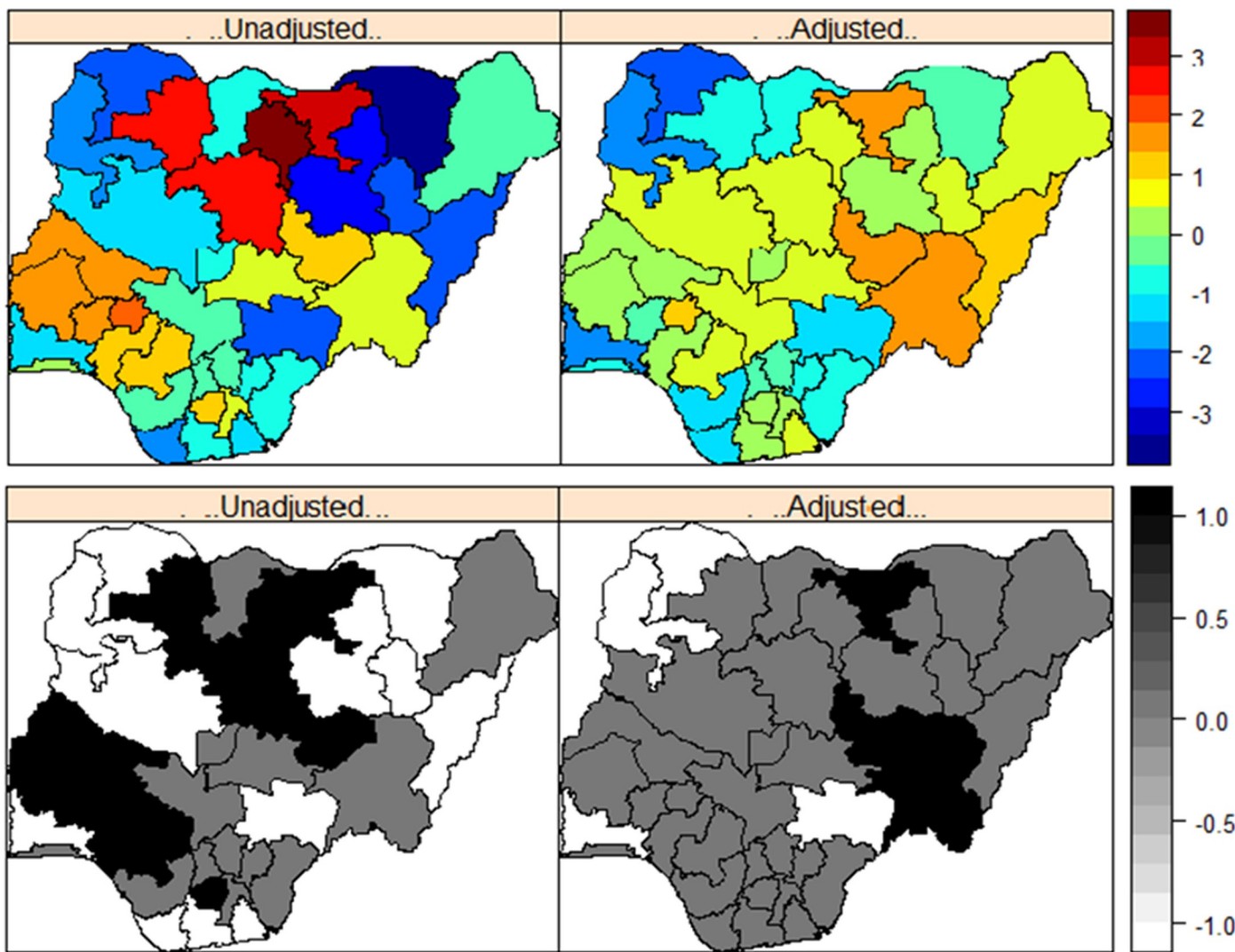

**Fig 4. Posterior estimates of the effects of a girl's likelihood of undergoing FGM/C in Nigeria based on the 2016 MICS dataset.** Mean (top panel). Significance maps of the posterior estimates based on 95% credible interval (bottom panel). Dark blue to dark red represent lowest to highest FGM/C risk states. Black, white and grey represent significantly high, low, and nonsignificant FGM/C risk states, respectively. Shapefile republished from DIVA-GIS database (https://www.diva-gis.org/) under a CC BY license, with permission from Global Administrative Areas (GADM), original copyright 2018.

they adjusted for variations due to survey differences ($m_2$) or not ($m_1$) suggesting that the methods used by the DHS and MICS were not significantly different in a statistical perspective. Similarly, Fig 5B shows that the *full* models $m_3$ and $m_4$ which did not account for potential interactions in space and time, showed no difference in fit regardless of whether variations due to different survey methods were accounted for ($m_4$) or not ($m_3$). However, the overall best fit was offered by the *full* model ($m_6$) which modelled space-time interactions as a random effect, that is, $\xi \sim N(0, \ \tau_\xi^2)$, suggesting that the changes in the spatial distribution of FGM/C prevalence among 0–14 years old girls in Nigeria happened independently in time or that the changes in prevalence over time happened regardless of the spatial location. Thus, for the pooled datasets only the results based on $m_6$ are presented and discussed further. However, for the purpose of comparisons, graphs and maps based on the unadjusted model which did not account for survey differences $m_1$ (simpler model) will be used where relevant.

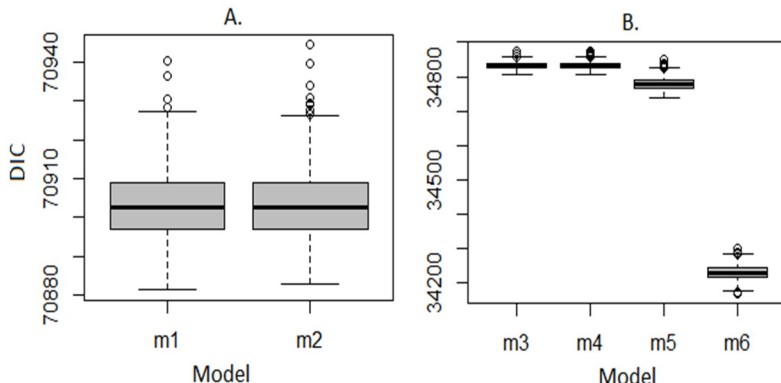

**Fig 5. Boxplot of the Deviance Information Criterion (DIC) for the six (6) models fitted to the pooled 2003 to 2016 dataset.** (A) Unadjusted model. (B) Adjusted model.

*Temporal trend*. A girl's likelihood of FGM/C varied over time and significantly peaked in 2008 and declined afterwards with significantly lowest likelihood found in 2011 (Fig 6). In addition, Fig 6 shows no significant progress in the reduction of the likelihood of FGM/C among 0–14 years old girls in Nigeria since 2011 suggesting that the practice is still being sustained despite the concerted abandonment efforts.

*Posterior Odds Ratio (POR)*. In Table 5, we present the results based on $m_6$ as posterior odds ratio (POR) and 95% credible intervals (95% CI). These results show the overall effects of the covariates across the years (2003 to 2016). We found that daughters of women who had no higher education were significantly more likely to be cut than other girls whose mothers had higher education. A girl whose mother had undergone FGM/C was more than 18 times more likely to be cut than a girl whose mother was not cut. Significantly lower risk of FGM/C was found among girls who lived in urban areas than in the rural areas. In support to earlier findings, we found that although the DHS data shows evidence of higher coverage, there were no

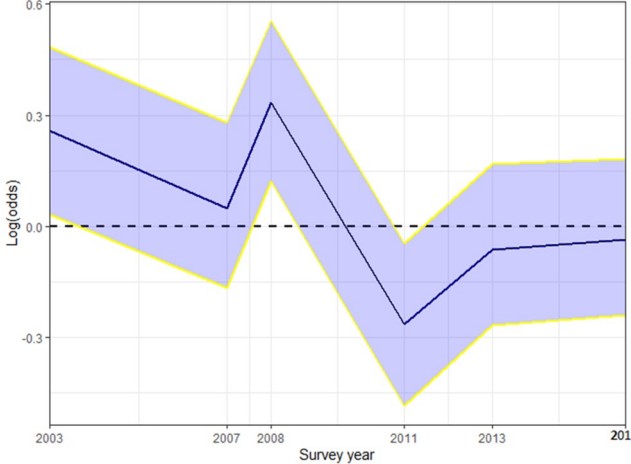

**Fig 6. Temporal trends in FGM/C among 0-14-year old girls in Nigeria ($m_3$) based on the pooled 2003 to 2016 dataset.** The blue lines are the posterior means; the yellow lines are the 95% credible intervals of the posterior estimates; and the blue band is the width of the 95% credible interval.

**Table 5. Posterior odds ratio estimates from the Bayesian geo-additive regression model (adjusted) fitted to the pooled 2003 to 2016 dataset.**

| Variable | | POR | CI | |
|---|---|---|---|---|
| Name | Remarks/level | | **2.5%** | **97.5%** |
| *Education* | Higher (ref) | 1.00 | - | - |
| | Primary | 2.35 | 2.313 | 2.504 |
| | Secondary | 1.928 | 1.91 | 2.01 |
| $fgm_{woman}$ | Not cut (ref) | 1.00 | - | - |
| | Cut | 18.153 | 17.923 | 19.125 |
| | Rural (ref) | 1.00 | - | - |
| *Residence* | Urban | 0.955 | 0.952 | 0.955 |
| *Year* | *See the graph, Figure* | - | - | - |
| *survey* | DHS (ref) | | | |
| | MICS | 0.877 | 0.724 | 1.082 |

POR- Posterior Odds Ration; CI- Credible Interval.

significant differences between the effects due to differences in survey methods suggesting that statistical evidence based on the DHS and MICS datasets are not statistically different.

*Effects of geographical location.* To address the question on the roles of geographical location on a girl's likelihood of undergoing FGM/C in Nigeria, we examine the posterior maps of the unadjusted and adjusted spatial effects. Fig 7 presents the posterior means (top panel) and the corresponding maps testing the statistical significance of the estimates based on 95% credible interval (bottom panel) for the unadjusted model ($m_1$;left) and the adjusted model ($m_6$;right). For these maps, dark blue to dark red correspond to lowest risk to highest risk states. While black, white and grey correspond to significantly high-risk states, significantly low risk states, and the states in which the effects of geographical location were not statistically significant. Based on $m_1$ higher likelihood of cutting existed among girls who lived in the North western states of Zamfara, Kaduna, Kano and Jigawa; South western states of Ekiti, Oyo, Osun, and Ondo; North central state of Kwara; South southern state of Edo and South eastern states of Ebonyi, Enugu and Imo. However, after taking account of the effects of other key covariates considered in the model (Table 5), the effects of geographical location in Osun state, Zamfara state and all the South eastern states became non-significant suggesting that over the years FGM/C have been sustained in the North western and South western states largely due to state-specific/local factors.

*Posterior estimates of FGM/C prevalence from 2003 to 2016.* In Fig 8, we present the predicted prevalence across the 36 states in Nigeria and the FCT based on the pooled data (Fig 8A), calculated as the back transformed posterior estimate of the linear predictor defined in (3), such that $p_k = \exp(\eta_k) / (1 + \exp(\eta_k))$ averaged over all girls in location (state) $k$, for. $k = 1,...,37$. We quantified the uncertainties in the estimation of the posterior prevalence using the deviance maps (Fig 8B) calculated as

$$D = 2 \sum \left\{ y_k \log\left(\frac{y_k}{\hat{\mu}}\right) + (n_k - y_k)\log\left(\frac{n_k - y_k}{n_k - \hat{\mu}}\right) \right\} \tag{9}$$

where $y_k$ and $\hat{\mu}$ are the observed and fitted values for the $k^{th}$ observation, respectively. Smaller values of $D$ indicate better fit.

These estimated prevalence maps are in agreement with the crude prevalence maps in Fig 3 with both consistently showing higher FGM/C prevalence in Southern Nigeria before 2008,

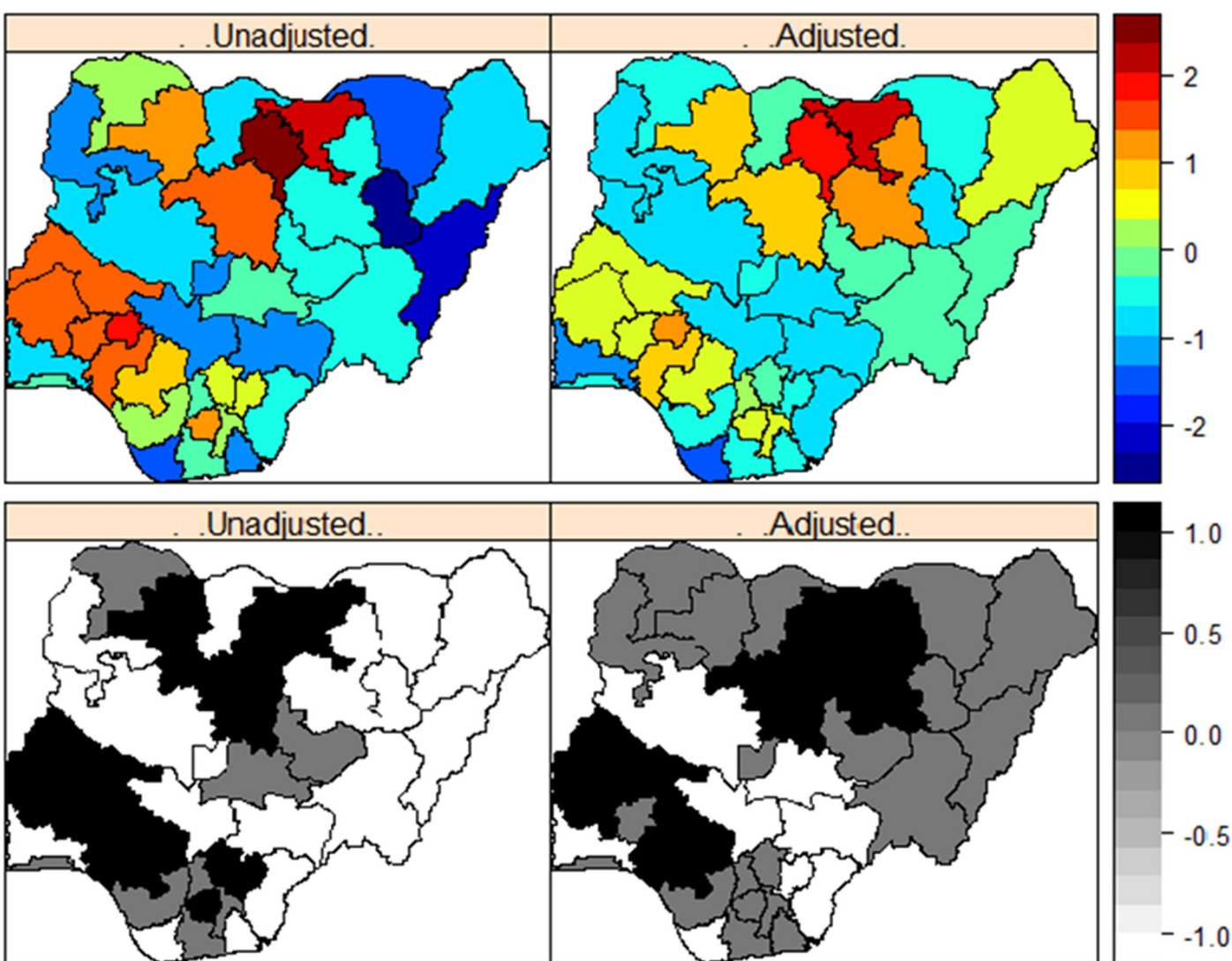

**Fig 7. Posterior estimates of the effects of a girl's likelihood of undergoing FGM/C in Nigeria based on the pooled 2003 to 2016 dataset.** Mean (top panel). Significance maps of the posterior estimates based on 95% credible interval (bottom panel). Dark blue to dark red represent lowest to highest FGM/C risk states. Black, white and grey represent significantly high, low, and nonsignificant FGM/C risk states, respectively. Shapefile republished from DIVA-GIS database (https://www. diva-gis.org/) under a CC BY license, with permission from Global Administrative Areas (GADM), original copyright 2018.

while a shift from lowest to highest prevalence occurred in the North western states since 2008 with Jigawa having the highest predicted posterior prevalence in 2016. All estimates showed moderate deviance values indicating high precision of the estimates. The advanced statistical methods employed in this study allowed us to better estimate FGM/C prevalence by accounting for errors in the response due to autocorrelations in space and time, and errors due to the different survey methods. Thus, we were able to estimate with very small standard error, the prevalence of FGM/C in states with no data by borrowing strengths from the information-rich neighbours. For example, we were able to estimate prevalence for Kano and Jigawa states in 2007 as well as the accurate prevalence for Katsina state in 2011 thus providing more reliable estimates of prevalence in time and space. Based on the 2016 estimated prevalence, the North

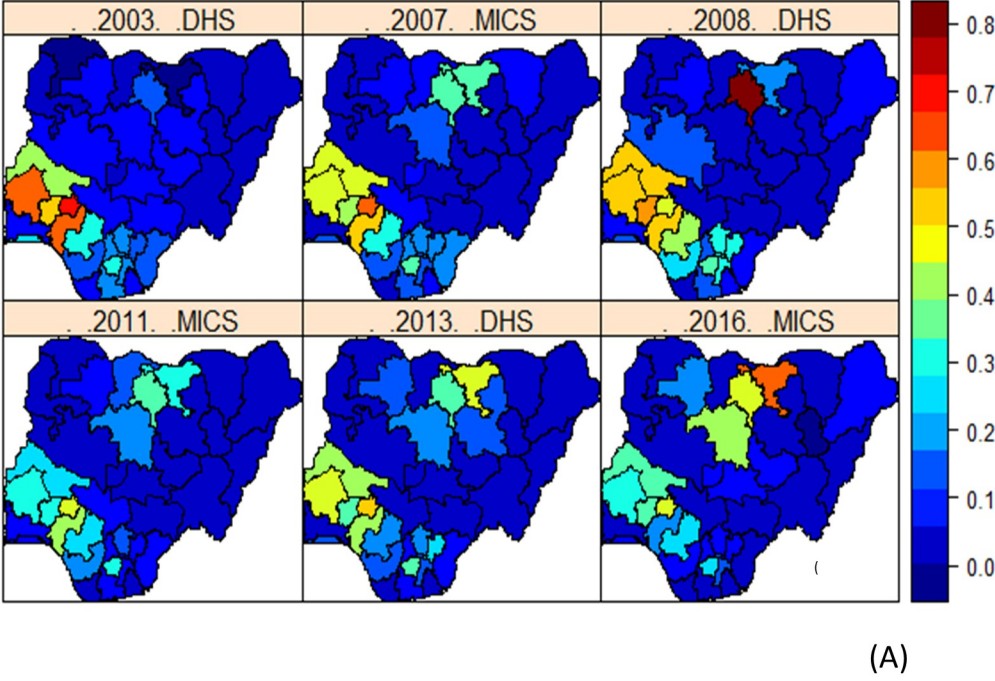

(A)

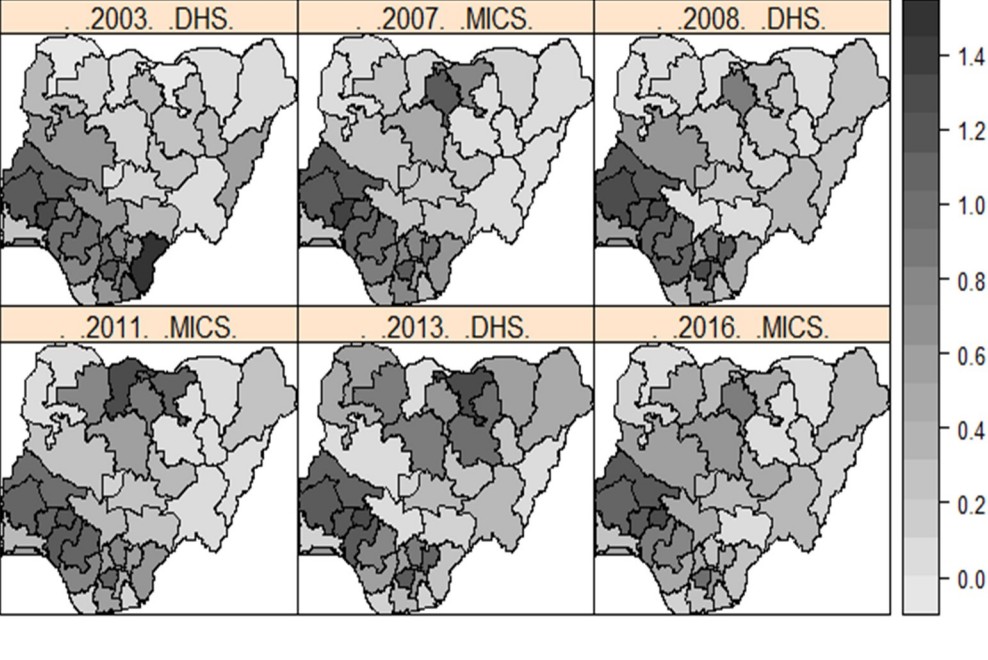

(B)

**Fig 8. Posterior estimates.** (A) Predicted prevalence of FGM/C among 0-14-year old girls in Nigeria from 2003 to 2016 based on $m_6$. (B) Corresponding deviance maps for the estimates. Dark blue to dark red implies lowest to highest prevalence. Shapefile republished from DIVA-GIS database (https://www.diva-gis.org/) under a CC BY license, with permission from Global Administrative Areas (GADM), original copyright 2018.

western states of Jigawa, Kaduna and Kano, and the South western state of Ekiti are identified as the hotspots where the practice is still rife.

*Non-linear effects of mother's age.* In Fig 9, we compare the effects of mother's age based on the pooled dataset from 2003–2016 (Fig 9A) to that based on the 2016 MICS dataset (Fig 9B). The combined effects of mother's age from 2003–2016 show that FGM/C prevalence was significantly high among girls whose mothers were more than 38 years old (9A). However, when the effects of mother's age were assessed using the recent 2016 MICS dataset, the effects of mother's age were no longer significant suggesting possible gains of interventions targeting older women.

## Discussion

This study evaluates changes in the prevalence of female genital mutilation/cutting (FGM/C) among 0–14 years old girls in Nigeria drawing upon data from multiple surveys. Specifically, we combined and analysed rich datasets from six (6) successive waves of nationally representative surveys comprising the Nigeria Demographic and Health Surveys (DHS) and Multiple Indicators Cluster Surveys (MICS) covering years 2003 to 2016/17. The objectives of the study were to provide more precise estimates of prevalence and better quantify the effects of the key determinants of FGM/C, and identify 'hotspots' where the practice is still rife to facilitate the development of more effective interventions. Four key questions were addressed: We examined the roles of key socio-economic and socio-demographic factors; the roles of social norms factors operationalised by a woman's FGM/C status, her beliefs, and her support for the continuation of the practice; the roles of a girl's geographical location in the persistence of the practice in Nigeria; and how changes in the practice varied over geographical locations and time. These questions were addressed using a combination of bivariate data analysis and a novel multivariate analysis which simultaneously adjusted for the inherent spatial and temporal autocorrelations within the data and the effects of other linear and non-linear covariates while accounting for potential variations due to the different survey methods, in a coherent Bayesian hierarchical spatio-temporal regression modelling framework.

Findings showed that FGM/C prevalence among 0-14-year-old girls in Nigeria varied geographically and temporally and has generally declined over the years in the South, but there has been a shift to the North since 2008. A girl was more likely to be cut if her mother was cut, supported FGM/C continuation, or had no higher education. Results from the bivariate analysis showed that most of the girls were cut before their 5[th] birthday with more than 80% of the cutting carried out by traditional circumcisers in 2016. Results based on a pooled 2003–2016/17 dataset showed that daughters of older women (38+ years) and girls who lived in rural areas were significantly more likely to be cut than the daughters of younger women and girls who lived in urban areas. However, evidence based on the 2016 MICS data alone shows that the effects of mother's age and place of residence (rural-urban) are no longer significant suggesting possible gains of interventions that targeted older women and women who lived in the rural areas. Higher likelihood of undergoing FGM/C were found among girls whose mothers believed that FGM/C was a religious obligation and prevents premarital sex. Furthermore, results suggest that the geographical location in which a girl lives matters. We found that over the years, the effects of geographical location were significantly high in most Southern and Northern states with girls who lived in the Northern states of Jigawa and Plateau having significantly higher likelihood of being cut in 2016. This implies that the practice of FGM/C in these states is being sustained largely due to some state-specific factors such as shared socio-cultural norms or due to their proximity to high prevalent neighbouring states (e.g., Kano and Jigawa states).

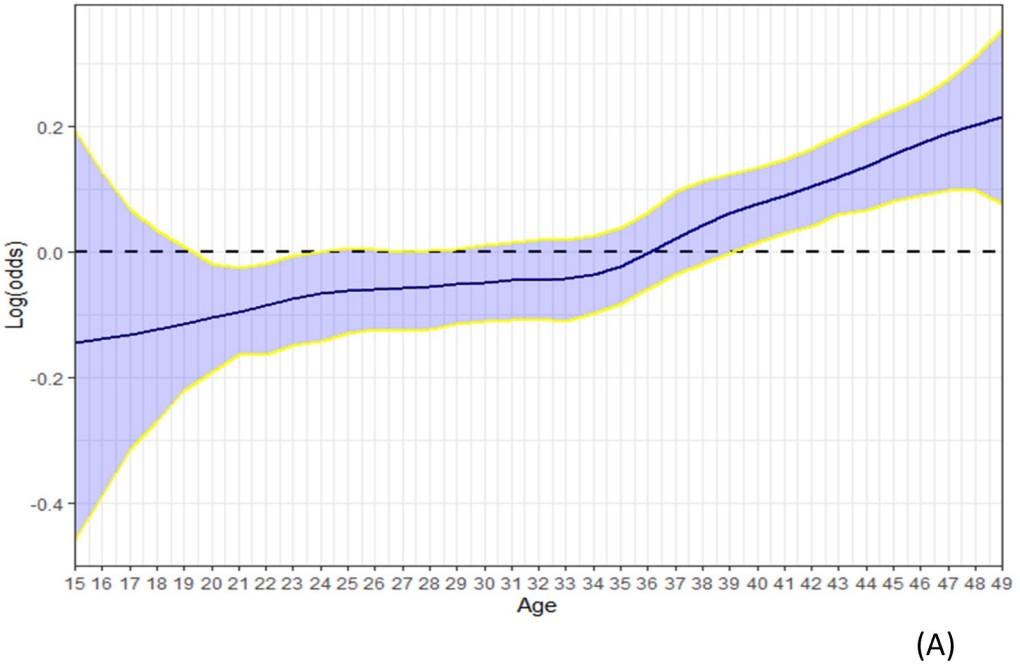

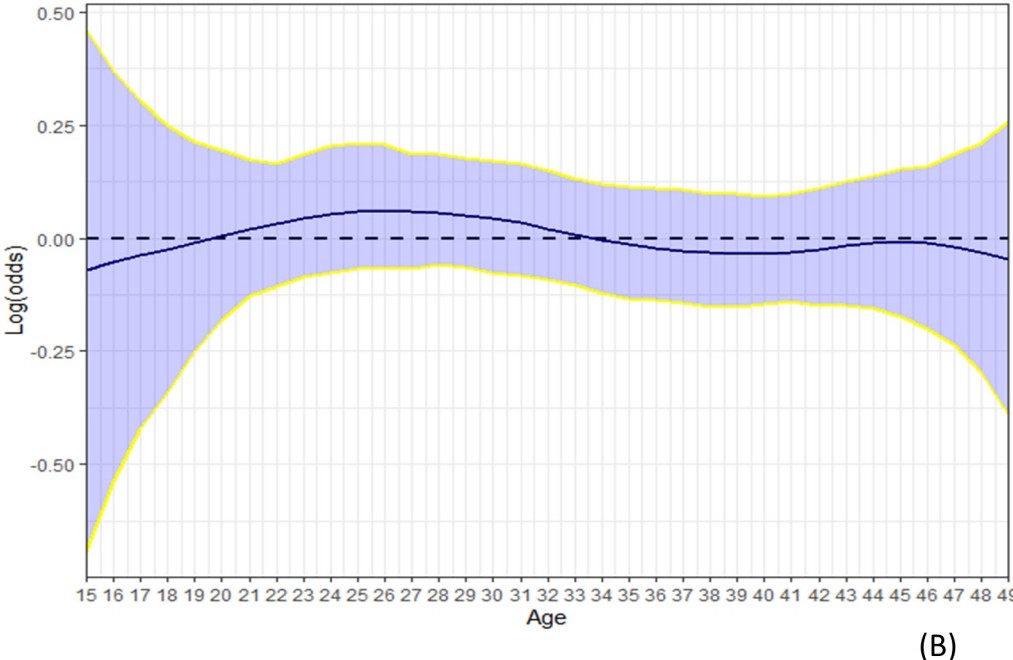

**Fig 9. Non-linear effects (log-odds) of mother's age on her daughter's likelihood of being cut based on (A) $m_6$ of the pooled 2003 to 2016 data and (B) Model B of the 2016 MICS data.** The blue lines are the posterior means; the yellow lines are the 95% credible intervals of the posterior estimates; and the blue band is the width of the 95% credible interval.

These results agreed with findings from previous studies [11, 17, 27–29, 45–47]. For example, it was found that education plays an important role in a mother's decision on whether her daughter will be cut or not [27], with mother's who had higher education less likely to cut their daughters in Nigeria [11]. This inverse relationship between a Nigerian woman's level of education and the likelihood of FGM/C was also reported in a scoping review of the determinants of FGM/C in Nigeria by Mberu in 2017 [17]. Also, a recent study by Cappa et al on the roles of parental attitudes on the likelihood of their daughter's FGM/C suggested that the opinions of both parents matter [45]. They found higher likelihood of FGM/C among African girls (including Nigeria) aged 0–14 whose mothers wanted FGM/C to continue supporting the assertion that FGM/C is a social norm. Similar results were found in the context of other countries [28, 46]. A study conducted in Senegal, Burkina Faso and Egypt by Farina and Ortensi found higher FGM/C prevalence among girls whose mothers had undergone FGM/C [46]. In the same spirit, studies conducted in the context of Kenya found that a woman's FGM/C status was key to her support for the continuation of the practice [28], and that daughters of women who were cut, women who supported FGM/C continuation and women who believed that FGM/C was a religious obligation, were more likely to be cut than their peers [29].

It should be noted, however, that majority of the studies mentioned above focused largely on the roles of individual- and community-level factors and the effects of the geographical locations in which a girl lives in Nigeria has only been considered in Kandala et al [27]. In addition, studies have mostly relied on single survey and single year data thus providing only a 'snapshot' evidence, while the need to understand the trends in the practice on a larger spatial and temporal scale has largely been overlooked. To the best of our knowledge, this is the first attempt to evaluate changes in FGM/C using data from multiple surveys over multiple number of years while accounting for uncertainties due to the different survey methods in a robust Bayesian hierarchical modelling framework. With this novel approach, we were able to exploit the strengths of each survey to examine the temporal trends in the practice, better quantify the effects of individual- and community-level factors, more accurately estimate prevalence and identify spatial patterns and 'hotspots' where the practice is still rife. The identified 'hotspots' included the North western states of Kaduna, Kano and Jigawa and the South western state of Ekiti thereby raising questions on the effectiveness of the intervention measures within these states.

The regional shifts in FGM/C prevalence are noteworthy as most of the intervention programmes have focused on high prevalence states in the South, with more emphasis laid on the education of the general public [17]. The North may therefore not have experienced the same FGM/C declines as the South because interventions have largely been focused on hotspots, which were thought to be only in the South based on previously available data examined cross-sectionally.

Nevertheless, the regional variances in FGM/C prevalence may stem from differences in the adoption of the anti-FGM/C laws. Although Nigeria enacted the 2015 Violence Against Persons (Prohibition) Act (2015 VAPP Act), which criminalises the practice of FGM/C in Nigeria, some Nigerian communities have continued to practice FGM/C unabatedly [10, 13]. As a federal law, the Act only applies in the Federal Capital Territory Abuja while states have to either adopt the law or create their own anti-FGM/C laws. Some states had their own anti-FGM/C laws prior to the passing of the VAPP Act. However, not all states have these laws in place and even where the law exists, the implementation may suffer setbacks due to lack of reporting probably due to family members involvement in the perpetration of the practice [1] and the limited presence of law enforcement agents, for example, in the rural areas.

Furthermore, the observed higher prevalence of FGM/C in the North since 2008 does not necessarily mean that the practice has recently been adopted. It is possible that data on FGM/C

in the North have been better collected in subsequent surveys. Thus, it might be that we are only just recently getting a clearer but not necessarily a new picture of FGM/C practice in the North. In addition, it is possible that these differences were a result of the different strategies used by the DHS and MICS in data collection before being standardised in 2010 [30]. However, we accounted for this potential source of error by adding an extra term *'rvey'* in the pooled data model to control for potential variations due to different survey methods, thus providing a clearer picture of the patterns of changes over the years.

In general, FGM/C is often viewed as a requirement that prepares a girl for marriage, inhibits urge for premarital sex and promotes marital fidelity [1]. Several theories have been advanced to explain why the practice of FGM/C persists despite sustained efforts geared towards its elimination. Prominent among these theories is the social norms theory which postulates that several decisions made by individuals are influenced by social norms within their community, and that FGM/C persists because those who practise it share mutual expectations (empirical and normative) with those in their reference group [20–23]. Thus, an individual's preferences and attitudes towards FGM/C ought to be understood within the context of the expectations and pressures that others in their reference group put on them [23]. As a result, an individual family is left to choose between the costs and benefits of adherence or non-adherence to social norms. Benefits may include enhanced marriageability and acceptance into peer social networks [21, 22] while costs may include social exclusion, persecution or ostracization. FGM/C is therefore considered a social norm because it is hard for an individual family to forgo it in isolation as long as most community members do not assent to abandoning it. Consequently, it has been suggested that programmes designed to promote FGM/C abandonment should approach it in terms of social norms theory [20–22].

Two major limitations of the study were identified. The first one is related to the use of self-reported data on FGM/C without clinical confirmation, which may lead to biased estimates. Although, a previous study on FGM/C in Tanzania suggests that some respondents make false self-reports that they have been cut due to social pressure to be circumcised [23], it is unlikely that this is always true. Nevertheless, for our purpose, it is expected that error due *to recall bias* will be minimal in that most women are able to confirm if their daughter was cut or not. A second limitation of this study is that we have considered only girls aged 0–14 years thus limiting the generalisation of the findings from the study. It is important to note that a girl who was not cut at age 14 may still be cut in future, hence this must be considered in interpreting our results noting that the FGM/C status of the girl at the time of the survey was not her final FGM/C status.

## Conclusion

Findings from this study have increased our understanding of the extent of success recorded in the fight against the scourge of FGM/C in Nigeria in the recent years. We have also gained further insights on where more resources should be mobilised in order to achieve a permanent abandonment in a social norm driven FGM/C high prevalence community. Such abandonments must be collectively adopted by the community or groups. Other members of the community would then be expected to conform with the *collectively* adopted change not to have their daughters cut with conformers socially rewarded while defaulters would possibly face social sanctions [22, 23]. A successful intervention programme must, therefore, be all-inclusive and involve partnership with both parents, local governments, policymakers, and community and religious leaders. Notable examples of such programmes are the 'FGM-Free Village Model' in Egypt and the Tostan Community Empowerment programme in Senegal. The advanced statistical approach utilised in this study is an important contribution to literature

and could be extended to other contexts where there is need to integrate data from multiple sources over large spatial and temporal scales to gain more statistical power.

## Acknowledgments

Authors would like to thank DHS and UNICEF for making the data used in the various analyses available.

## Author Contributions

**Conceptualization:** Chibuzor Christopher Nnanatu, Ngianga-Bakwin Kandala.

**Data curation:** Chibuzor Christopher Nnanatu, Glory Atilola, Lubanzadio Mavatikua, Zhuzhi Moore.

**Formal analysis:** Chibuzor Christopher Nnanatu, Glory Atilola, Zhuzhi Moore.

**Methodology:** Chibuzor Christopher Nnanatu.

**Project administration:** Lubanzadio Mavatikua.

**Software:** Chibuzor Christopher Nnanatu.

**Supervision:** Ngianga-Bakwin Kandala.

**Writing – original draft:** Chibuzor Christopher Nnanatu.

**Writing – review & editing:** Chibuzor Christopher Nnanatu, Glory Atilola, Paul Komba, Lubanzadio Mavatikua, Zhuzhi Moore, Dennis Matanda, Otibho Obianwu, Ngianga-Bakwin Kandala.

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
