## [Decision Letter · Decision Letter 0]

3 Dec 2020

PONE-D-20-16745

Evaluating changes in the prevalence of female genital mutilation/cutting among 0-14 years old girls in Nigeria using data from multiple surveys: A novel Bayesian hierarchical spatio-temporal model

PLOS ONE

Dear Dr. Nnanatu,

Thank you for submitting your manuscript to PLOS ONE. After careful consideration, we feel that it has merit but does not fully meet PLOS ONE’s publication criteria as it currently stands. Therefore, we invite you to submit a revised version of the manuscript that addresses the points raised during the review process.

We look forward to receiving your revised manuscript.

Kind regards,

Kannan Navaneetham, PhD

Academic Editor

PLOS ONE

Journal Requirements:

2. Please ensure that you in your Methods section report the  question used to derive data (both outcome variables and explanatory variables), and specify how variables were defined and categorised. Please also ensure that you discuss missing data and imputation.

3.We note that [Figure(s) 1, 3, 4, 7 and 8] in your submission contain map/satellite images which may be copyrighted. All PLOS content is published under the Creative Commons Attribution License (CC BY 4.0), which means that the manuscript, images, and Supporting Information files will be freely available online, and any third party is permitted to access, download, copy, distribute, and use these materials in any way, even commercially, with proper attribution. For these reasons, we cannot publish previously copyrighted maps or satellite images created using proprietary data, such as Google software (Google Maps, Street View, and Earth). For more information, see our copyright guidelines: http://journals.plos.org/plosone/s/licenses-and-copyright.

1.    You may seek permission from the original copyright holder of Figure(s) [1, 3, 4, 7 and 8] to publish the content specifically under the CC BY 4.0 license. 

Reviewers' comments:

Reviewer's Responses to Questions

**Comments to the Author**

1. Is the manuscript technically sound, and do the data support the conclusions?

Reviewer #1: Yes

Reviewer #2: Yes

2. Has the statistical analysis been performed appropriately and rigorously? 

Reviewer #1: Yes

Reviewer #2: I Don't Know

3. Have the authors made all data underlying the findings in their manuscript fully available?

Reviewer #1: Yes

Reviewer #2: Yes

4. Is the manuscript presented in an intelligible fashion and written in standard English?

Reviewer #1: Yes

Reviewer #2: Yes

5. Review Comments to the Author

Reviewer #1: Generally:

1. what is the reason combined different year data set?

2.why not use the informative prior?

3.how to handle the missing values? please explain the algorithm to solve these this problems.

4. why not separate these discussion and conclusion?

Reviewer #2: • Well written

• Please refer to the Fig 1. The labelling of the diagram is wrong as the States in the South East of Nigeria were labelled as South South and vice versa.

• Kindly go through all the statistical reports to be sure there is no mix-up of data reported for the various geographical zones and if any, please make appropriate corrections throughout the manuscript.

6. PLOS authors have the option to publish the peer review history of their article (what does this mean?). If published, this will include your full peer review and any attached files.

Reviewer #1: No

Reviewer #2: No

---

## [Author Response · Author response to Decision Letter 0]

16 Jan 2021

Included in the attached document called "Response to Reviewers"

---

## [Editor Report · Decision Letter 1]

25 Jan 2021

Evaluating changes in the prevalence of female genital mutilation/cutting among 0-14 years old girls in Nigeria using data from multiple surveys: A novel Bayesian hierarchical spatio-temporal model

PONE-D-20-16745R1

Dear Dr. Nnanatu,

We’re pleased to inform you that your manuscript has been judged scientifically suitable for publication and will be formally accepted for publication once it meets all outstanding technical requirements.

Kind regards,

Kannan Navaneetham, PhD

Academic Editor

PLOS ONE
---

## [Editor Report · Acceptance letter]

27 Jan 2021

PONE-D-20-16745R1 

Evaluating changes in the prevalence of female genital mutilation/cutting among 0-14 years old girls in Nigeria using data from multiple surveys: A novel Bayesian hierarchical spatio-temporal model 

Dear Dr. Nnanatu:

I'm pleased to inform you that your manuscript has been deemed suitable for publication in PLOS ONE. Congratulations! Your manuscript is now with our production department. 

Kind regards, 

on behalf of

Professor Kannan Navaneetham 

Academic Editor

PLOS ONE